# The AIDSS Module for Data Acquisition in Crisis Situations and Environmental Protection

**DOI:** 10.3390/s20051267

**Published:** 2020-02-26

**Authors:** Andrija Krtalić, Milan Bajić, Tamara Ivelja, Ivan Racetin

**Affiliations:** 1Faculty of Geodesy, University of Zagreb, 10000 Zagreb, Croatia; 2Scientific Council, HCR–Centre for Testing, Development and Training, 10000 Zagreb, Croatia; milan.bajic1@gmail.com; 3Zagreb University of Applied Sciences, 10000 Zagreb, Croatia; tivelja@tvz.hr; 4Faculty of Civil Engineering, Architecture and Geodesy, University of Split, 21000 Split, Croatia; racetin@gradst.hr

**Keywords:** multi-sensors system, aerial data acquisition, sensors, platforms, remote sensing

## Abstract

The Toolbox implementation for removal of antipersonnel mines, submunitions and unexploded ordnance (TIRAMISU) Advanced Intelligence Decision Support System is an operational system proposed to Mine Action Centres worldwide for conducting non-technical surveys in humanitarian demining. The system consists of three modules, one of which is the module for data acquisition introduced and described in this study. The module has been designed, produced, improved, used and operationally tested and validated on several platforms (helicopters, remotely piloted aircraft systems (RPAS) and a blimp), with various sensors and acquisition units (Global Positioning System (GPS) and inertial measurement unit) in a variety of combinations for additional data acquisition from deep inside a suspected hazardous area. For the purposes of aerial data acquisition over a suspected hazardous area, the use of multiple sensors such as visible digital cameras and multi-spectral visible, near infrared (VNIR), hyperspectral VNIR and thermal infrared sensors are of benefit, because they display the scene in different ways. Off-the-shelf equipment and software were mostly used, but some specific equipment, such as sensor pods, was developed and also some software solutions for data acquisition and pre-processing (transforming hyperspectral line scanner data into hyperspectral images, and producing hyperspectral cubes). The technical stability and robustness of the module were confirmed by operationally testing and evaluating the systems on the aforementioned platforms and missions in several actual suspected hazardous areas in Croatia and Bosnia and Herzegovina, between 2001 and 2015.

## 1. Introduction

After the end of the Homeland War in 1995, the Republic of Croatia faced a huge mine problem. A great deal of its territory (about 10.5%; [1]) was suspected hazardous areas. “A suspected hazardous area is an area where there is reasonable suspicion of mine/explosive remnants of war contamination on the basis of indirect evidence of the presence of mines/explosive remnants of war” [2]. During the post-war period, concern increased regarding the number of landmine injuries suffered by innocent civilians. The civilian approach to demining (humanitarian mine action) differs from the military approach and begins when the conflict stops. It is called mine action [2]. The goal of integrated mine action is to return previously mined land to the community for use and covers a far wider scope of activities than simply mine clearance. It includes mine awareness and risk-reduction education, minefield surveying, mapping, marking and clearance, victim assistance, including rehabilitation and reintegration, and advocacy to stigmatize the use of landmines and support a total ban on anti-personnel landmines [3]. Mine action ultimately aims at a 100% clearance rate of land mines and dangerous explosive objects [3]. To plan and implement demining projects successfully, it is necessary to know minefield locations. Information-gathering techniques, such as interviewing returnees, general mine action assessment [4], technical surveys [5], analytical evaluation of military maps or reading the biographies of military commanders, provide a good insight into the mine situation [6,7]. However, as noted in [3] these are long, expensive processes that ultimately do not provide enough accurate information. The mine action process must be accelerated in order to identify mined areas quickly, avoid accidents, and assign demining priorities. This requires a quick, low-risk, cost-effective tool for surveying suspected hazardous areas and producing maps with indicators of the mine presence and absence [8,9,10] to define suspected hazardous area boundaries. For example, in the Republic of Croatia, humanitarian mine campaigns have shown that only 10% of suspected hazardous areas are mine-affected [11]. It is almost as important to identify areas not affected by landmines, for the purpose of reducing an already defined suspected hazardous area [12]. 

To this end, the Advanced Intelligence Decision Support System (AIDSS) [13,14] based on visible digital cameras, multi-spectral and hyperspectral visible, near infrared (VNIR), and thermal infrared sensors, has been designed, produced, operationally validated and implemented in the Republic of Croatia and Bosnia and Herzegovina. AIDSS is the result of research conducted within seven international and domestic scientific projects: ARC [15], SMART [12], System for Multi-sensor Airborne Reconnaissance and Surveillance in Crisis Situations and Environmental Protection [16], Deployment of the Decision Support System for Mine Suspected Area Reduction [17], Deployment of the Advanced Intelligence Decision Support System for Mine Suspected Area Reduction in Bosnia and Herzegovina [18], and Toolbox implementation for removal of antipersonnel mines, submunitions and unexploded ordnance (TIRAMISU) [19], which aimed to implement airborne and satellite-borne remote sensing for non-technical survey [10,20] in humanitarian mine campaigns and crisis situations. AIDSS is a modular tool with a module for aerial data acquisition [14]. 

The goal of the AIDSS module for aerial data acquisition, or any similar multi-sensor imaging system, is to provide usable images for extracting information, detecting and identifying objects and features based on image (geometric and spectral) characteristics, and classification as shown in [12] and [13]. Multi-sensor imaging systems as cited in [21,22,23,24,25] allow the implementation of multi-sensor data fusion [26,27], the results of which can reveal certain objects indirectly, as shown in [28]. Multi-sensor fusion deals with a combination of complementary, sometimes competing sensor data, in a reliable estimate of the environment, to achieve an outcome that is better than the sum of its parts, [29,30,31], to achieve inferences that are not feasible from each sensor operating separately. Advances in the development of sensor technology are insufficient without the use of multi-sensor fusion techniques [32]. Since sensors of different types integrated in the system have their own limitations and perceptive uncertainties, an appropriate data fusion approach is expected to reduce overall sensor uncertainties and increase the accuracy of system performance [33]. The crucial thematic framework which assures operational success and the outcome values of multisensory imaging and fusion for the purposes of anti-mine operations is the detailed identification of the mined scene, with the working title of analytical assessment, defining the so-called strong indicators of mine levels, introduced in 2001 and developed up to 2015 [6,7]. These paradigms have assured the selection and adaptation of sensory techniques, a regime for recording situations in suspected hazardous areas, and a new type of outcome, called “office virtual reconstruction of former battlefields”.

Data fusion in AIDSS is not performed within the data acquisition module, but after processing the collected materials in the third module and, therefore it will not be explained any further in this paper. Here, it is important to stress that data fusion is not performed in real time, as with other multi-sensory systems, for example [34] or [21]. Therefore, it is important to define the purpose of multi-sensory imaging well, in order to select optimal sensors and adequate platforms for such a system.

A thorough review of satellite and airborne sensors for remote-sensing-based detection of minefields and landmines can be found in [11], focusing on multi-temporal aerial photographs and satellite images. The paper presents a good analysis of the structure and composition of minefields and patterns that can be obtained for minefield detection. On the other hand, [32,35,36] focus on some of the most common ‘direct’ remote sensing technologies in landmine detection, defining ‘direct’ as mine clearance per se (a technology used in technical surveys and actual demining processes). The demand for detailed information from inside suspected hazardous areas has increased markedly throughout the world. Therefore, it is very important to develop new remote-sensing techniques that allow for the direct measurement of common situations in suspected hazardous areas. Although public satellite imagery is available (e.g., Sentinel, Landsat), low ground sample distance (GSD, 10 m for Sentinel-2 and 30 m for Landsat satellite imagery) is an obstacle to interpreting and extracting information with the required accuracy. On the other hand, commercial satellites provide better GSD (≈0.5 m from WordView and GeoEye satellite imagery) and better insights into the scene. Airborne imagery resolves the low-GSD problem of satellite imagery and, with better GSD, provides adequate spectral bands for photogrammetric and remote-sensing methods. Remotely piloted aircraft systems (RPAS) provide special advantages over other platforms, in particular high GSD, and more economical preparations for performing and collecting data in smaller areas. Examples of RPAS use in landmine detection can be found in [37,38,39], and indicators of mine presence detection in [40,41].

This paper presents an overview of an imaging system operating in the visible, near infrared and LWIR range of the electromagnetic spectrum according to defined needs, particularly humanitarian mine action, and which has also been tested to collect data on the quality of standing water, oil pollution in the Adriatic and fire monitoring [16,42]. The results of this system are used as inputs in data fusion and the production of mine danger maps [6,7,10,43].

## 2. Equipment

### 2.1. Advanced Intelligence Decision Support System (AIDSS) Module for Aerial Data Acquisition

AIDSS is not a mine detector. It is a set of tools and methods, advanced and integrated into one effective system, and based on SMART methodology [44] for use by experienced operators, experts in remote sensing and experts from Mine Action Centres, to help suspected hazardous area reduction using remote sensing data and expert knowledge [14]. In the SMART project [45], many useful tools for aerial non-technical surveys in humanitarian mine action were developed and tested, but within the actual project and following it, they were integrated and implemented operationally. AIDSS was developed due to actual demands for help in removing mines quickly in the Republic of Croatia. AIDSS is a complex system (Figure 1) to support decisions on defining suspected hazardous areas. It consists of three modules:module for the analytical assessment of mine information system data;module for data acquisition (multi-sensor imaging system);module for data pre-processing and processing.

The modules can be used together or individually. Input includes data from the mine information system, expert knowledge, and airborne, satellite and contextual data (Figure 1). AIDSS is a unique mine action technology that provides a successfully operational system combining remote sensing with advanced intelligence methodology. It is a validated operational solution for non-technical surveys in humanitarian mine actions [13] proposed to mine action centres worldwide, because it is adaptable to specific terrains and situations. The outcomes of this system are successfully detected and confirmed geographical positions of indicators of mine presence and indicators of mine absence, a better (re)definition of a suspected hazardous area, and thematic maps (mine danger maps) [6]. Therefore, the AIDSS module for data acquisition is a very important part of the whole system, because the processing of collected data should provide useful results, within the AIDSS methodology, for decision-making on suspected hazardous area re-definition.

The goal of the AIDSS module for data acquisition for humanitarian mine action is to collect information about the current situation within a suspected hazardous area or munitions depots destroyed by explosion, that is, information on the locations of remains of fortification objects in the area. Fortification objects are strong indicators of mine presence, such as trenches, bunkers, artillery tool stores, personnel shelters, altered forest boundaries, objects that are not currently used but were used before the conflict, and the remains of military equipment. 

Suspected hazardous area in the Republic of Croatia are usually not compact areas with regular borders, but larger and smaller fragmented areas. For this reason, helicopters and RPAS were chosen as platforms, and a blimp was also tested as a potential platform. Helicopters are mobile platforms which can change direction and require low minimum speeds for stable flight (Mi-8 ≈ 120 km/h, Bell-206 ≈ 70 km/h) in comparison with airplanes. Helicopter flights require shorter times, particularly in maneuvering from one site and set of images to another. Helicopters can also fly lower (≈200 m above the ground) for minimum speed flight stability, and this directly affects the size of GSD on images. For an even better view of details in small areas of suspected hazardous areas, or in parts recorded from helicopters where GSD was inadequate to detect individual indicators of mine presence, RPAS was used as a multi-sensor system platform. These platforms can fly low and hover over areas of interest within a suspected hazardous area.

Research and development to create AIDSS has responded to real demands from humanitarian mine action experts in Croatia. It was done using prototyping (spiral) methodology rather than waterfall methodology, and the statement of needs was defined before each step was defined, followed by state of the art, and ending with gap-filling requirements. An initial version of the system (prototype) was developed that was modified according to the needs of each project and tested on each mission. 

The module ensured the stability and reliability of data acquisition on each platform. The technical stability and the robustness of the system has been confirmed by tests and evaluations (based on the behavior of the system during data acquisition over areas of interest) on different platforms and missions in the Republic of Croatia and Bosnia and Herzegovina in the periods mentioned.

The AIDSS module for data acquisition consists of:Visible digital cameras;Multi-spectral VNIR sensor;Thermal infrared sensor;Hyperspectral VNIR sensor;Sensors for navigation and positioning (Global Positioning Systems (GPSs) and inertial measurement unit);Module control system;Power supply;Platforms (helicopters, RPAS, blimp);Operator.

### 2.2. Sensors

The passive sensors may detect natural electromagnetic energy that is reflected or emitted by the observed object. Two main categories of passive camera systems can be distinguished: frame cameras and line-scanner imaging systems [46]. Various types of both (multi-spectral VNIR: Fuji FinePix, Canon 5D, Nikon D90, SONY α6000, DuncanTech MS3100, Redlake MS4100; hyperspectral VNIR: ImSpector V9/PixelFly, UHD; panchromatic LWIR: THV 1000, Photon 320) were investigated and used in AIDSS’s module for aerial data acquisition (Table 1). A wealth of experience was gained, and random selection was narrowed down while refuting the frequently expressed claim, “the more data from different sensors, the greater the probability of success” [36]. No single technology has the capability to detect and recognize a variety of indicators of mine presence under all circumstances [32]. Most developed technologies and techniques are complex and/or expensive. Many are promising, but none has the sensitivity, size, weight, manufacturability and price range required for humanitarian mine action [32,36]. The goal of the AIDSS module for data acquisition is to collect information about the current situation within a suspected hazardous area or munitions depot destroyed by a explosion, that is, information on the locations of remains of fortification objects in that area. Sensors for the AIDSS module for data acquisition were selected according to the above requirements (Table 1). Indicators of mine presence can be detected and isolated on digital images using some of the methods for processing digital images described in [47,48], or by methods of object-oriented identification of linear objects based on presuppositions regarding their geometric and radiometric features and use of various filters to emphasize them [49] or [50]. Isolating indicators of mine presence on hyperspectral images is done via their spectral characteristics, as shown in [51,52,53].

#### 2.2.1. Visible Digital Cameras

Visible digital matrix cameras in the present constellation of sensors are the Nikon D90 and SONY α6000, which collect information in the visible part of the spectrum, in 3 spectral bands from 400–700 nm (Table 1). The main cameras’ technical specifications are listed in Table 1. The Nikon D90 has high signal-to-noise components and design, and delivers exceptional performance, even at high ISO setting and GPS unit to provide automatic real-time geotagging [54]. It was included in the sensor system for the AIDSS data acquisition module due to its robustness and technical characteristics. The Sony α6000 E-mount camera is a compact, light-weight camera with interchangeable lenses. The α6000 compares favorably to bulkier, heavier DSLRs, and with interchangeable lenses, manual controls and more [55]. This is very important when selecting sensors for an aerial multi-sensor imaging system, as the payload can be reduced, which is important when constructing supports for an unmanned aerial vehicle (UAV) and calculating endurance without reducing the quality of the images collected. These parameters and the camera’s ability to freeze a subject at 11 fps [55] for shots that capture the exact moment or object of interest, were decisive factors in its selection. These images were used to gain a better insight into the situation in the scene and detect indicators of mine presence according to geometric characteristics (trenches, bunkers, various types of shelter).

#### 2.2.2. Multi-Spectral Visible, Near-Infrared (VNIR) Sensor

The Redlake MS-4100 is a multi-spectral VNIR optical matrix camera with 3 separate CCD sensors and is available in two spectral configurations. The first is RGB for high quality colour imaging, and the second is colour-infrared for multi-spectral applications (4 spectral bands from 400–1000 nm). Standard colour-infrared imaging (CIR) uses red, green and near-infrared bands approximating Landsat satellite bands. The maximum frame of the MS-4100 is 10 fps with a pixel clock rate of 25 MHz and bit depth of 12 bits [56].

#### 2.2.3. Thermal Infrared Sensor

A FLIR Photon 320 [57] LWIR (8–14 µm) uncooled microbolometer camera was used for collecting thermal images from 2008 until 2016. The Photon 320 had a 50 mm lens providing a 14° horizontal and 11° vertical field of view and acquired image frames of 324 × 256 pixels as raw 14-bit digital numbers at the rate of 9 Hz. Image sequences from the camera were converted into ethernet data packets by the FLIR Ethernet module and this data was then stored on a computer on the used platform (helicopters or RPASs). The system time of the computer was set to GPS time prior to flight, so that the thermal data files could be synchronized with GPS log files. Panchromatic LWIR Photon 320 sensors have poorer GSD than the visible digital cameras, however, they are used to detect indicators of mine presence via their thermal characteristics.

#### 2.2.4. Hyperspectral VNIR Sensor

Conventional commercial spectrometers or spectrophotometers are usually able to measure the optical spectrum from a specified surface area at one point [58,59]. This is done either with one detector scanning the spectrum in narrow wavelength bands, or with an array detector, in which case all the spectral components are acquired at once. If the spectrum is to be measured at several spatial locations of the specified surface, the target under examination or the measuring instrument must be mechanically scanned. In Section 3.3., the procedure will be shown for creating a hyperspectral cube from sequential, continuous samples. 

An imaging spectrometer instrument, based on an imaging spectrograph like the ImSpector V9, is “an instrument capable of simultaneously measuring the optical spectrum components and the spatial location of an object surface” [60]. The ImSpector V9 hyperspectral line scanner is a direct sight imaging spectrograph and was combined with a PCO PixelFly high performance digital 12-bit CCD monochrome matrix camera [61] to form a geometric sensor model-imaging spectrograph constructed for this particular module. The PixelFly matrix camera with a scan area of 8.6 × 6.9 mm and effective pixels of 1280 (H) × 1080 (V) consists of an ultra-compact camera head, which either connects to a standard PCI or a compact PCI board via a high-speed serial data link. The available exposure times range from 5 μs to 65 s [61].

#### 2.2.5. Sensors for Navigation and Positioning

The system for navigation, determining the position and orientation of the system in space, consists of a single-frequency GPS device integrated with an inertial measurement unit (IMU) iVRU-RSSC by iMAR GmbH, and additional GPS units arranged in or on the platform. IVRU-RSSC is a triple-axis inertial system with three mutually perpendicular MIL-MEMS gyroscopes for determining the angle elements of the spatial orientation of the sensor, and 3 MEMS-accelerometers to determine the acceleration components along all three axes. The device has an integrated microprocessor for 16-bit digitalization of data from the sensor and deviation correction, to improve the accuracy of all measured elements. The GPS data are used primarily to correct navigation solutions acquired from the inertial system. Although the internal IMU working rate is 200 Hz, for this purpose, the elements of current position and sensor orientation in relation to the referential WGS84 coordinate system were stored with a frequency of 20 Hz. This allowed high-quality fluctuation and raw element bias correction using the internal processor. On the other hand, the volume of redundant data was reduced, and further processing was made easier, since the frequency of imaging of the ImSpector V9 sensor is 10 Hz. The elements of the external orientation of the platform related to the WGS84 coordinate system are expressed in ellipsoidal coordinates φ, λ and h on the GRS80 ellipsoid. Alongside the IMU, a separate GPS device was used with an aerial to synchronize the computer time with GPS time (UTC).

### 2.3. Module Control System 

The image capture rate was controlled by an operator inside the helicopter or remotely from the RPAS flight control board. A special command desk was made to gather data from the Mi-8 helicopter (Figure 2) to manipulate the module. Until 2009, desk-top computers were used to manage sensors in helicopter platforms. Due to vibration (particularly in the Mi-8) the computers crashed from time to time and communication with the sensors was lost. So, the desktop computer has been replaced by industrial controllers with solid-state drives (SSD) which are more robust and resistant to vibration. Small, custom-made computers were used for the sensor control on RPAS platforms. The Nikon and Sony cameras were operated in shutter priority mode (a fast shutter speed was required to minimize motion blur), in which the desired shutter speed (depending on the altitude and speed of the platform and light) was set before flight and the exposure was adjusted automatically by varying the aperture. Images from the Sony α6000 camera were captured in RAW format and stored on the memory card in the camera for post-flight download.

DuncanTech MS-3100 and the ImSector V9 + Pixelfly hyperspectral VNIR system were managed using the RECORDER program, which was developed and produced specially for this purpose within the ARC project. It includes changing sensor parameters before and during flight, individual images exported to standard TIFF format, and the metadata for each image are stored in the corresponding table (time of recording, GPS and IMU data). Based on these data, synchronization with GPS and IMU data is performed.

### 2.4. Module Power Supply

The acquisition systems on board the Mi-8 and Bell-206 helicopters used their own sources of electric power, but on the Mi-8, could also use electricity from the helicopter’s power supply system. Previous practice has shown certain problems when connecting to the helicopter’s electrical system (a special type of connector is needed, equipment is subject to obsolescence, and installation is impossible on helicopters which are over 50 years old). Therefore, it is essential to ensure the independence of the system’s power supply from the platform’s power supply. The variety of electrical power sources used in the helicopter also decreases the operational availability of the system. A continuous, stable electricity supply for aerial data acquisition is mandatory. It is also essential for the stable, continuous operation of the module during flight. The major obstacle in this regard was the need to convert electricity from 28–30 V direct current (DC) to 220 V alternating current (AC) 50 Hz (in the initial variant of the system). Therefore, the power supply for the module for aerial data acquisition was re-designed in 2012. This was done by replacing the desktop PC with two industrial controllers (one was already embedded in 2009) and a monitor operated by 24 V (controllers) and 12 V (monitor) DC. The instruments and equipment for the modules on the helicopters were powered by an independent power supply consisting of two large batteries (210 Ah, 75 kg each) in the Mi-8 helicopter and one in the Bell-206 helicopter. Small batteries were used to power a module installed on the RPASs.

### 2.5. Platforms

Mi-8, Bell-206 and Gazela helicopters, RPAS X8 MK and RPAS 8 ZERO and a blimp (Figure 3) were operationally tested over the sites in question.

#### 2.5.1. Helicopters

The first platform used in 2003 for an aerial multisensory system during the trial data acquisition on the current situation in the suspected hazardous areas, as part of project ARC, was a Bell-206 helicopter. The crew comprised the pilot, co-pilot and two systems operators. Based on the results and experience gained, the first AIDSS module for aerial images acquisition was planned, made and used on an Mi-8 military helicopter (Figure 3a) and a Bell-206 (Figure 3c). The crew of the Mi-8 comprised the pilot, co-pilot, technician, systems operator, navigator and mission leader (it is possible to increase the number of people involved in each segment of the system). In order to extend the system and increase the amount of equipment carried, the current crew consists of three members, the pilot, co-pilot and systems operator. The maximum endurance (along with an additional tank of fuel inside the helicopter) of the Mi-8 is 4 h 15 min, and of the Bell-206, 2 h 15 min. So, if the endurance is at least 3 h for the Mi-8 (1 h 15 min for the Bell-206) it is theoretically possible to acquire high-resolution multi-sensor imagery from an area of about 45 sq. km (≈ 25 sq. km for the Bell-206) per flight over flat terrain, at a relative height of 1000 m in one continuous sequence, without loops. The Gazelle helicopter (Figure 3f) was used in Bosnia and Herzegovina in 2014, and can transport up to five passengers, with 500 kg of internal space in the rear of the cabin. There was an advantage when using the module, as the helicopter already had an opening in the floor, which made it easier to install the equipment without compromising the flying ability of the platform. 

#### 2.5.2. Remotely Piloted Aircraft Systems (RPAS) and Blimp

Since the suspected hazardous areas were not compact, but comprised a large number of scattered areas, and in the interests of economizing resources for data acquisition on such areas, several RPAS platforms were tested for the specific task of collecting high spatial resolution hyperspectral data. For this purpose, the TIRAMISU Light Hyper Spectral Imaging System (T-LHSIS) for aerial data acquisition was developed, installed and tested on two RPAS platforms, X8 MK and 8 ZERO, and a blimp [53]. Both multirotor RPASs tested fall into the category of small RPAS s with take-off mass below 10 kg. X8 MK was tested on several occasions at several locations. The main problem in this project [62] was to provide RPAS for a payload of about 4 kg with several additional requirements related to collecting hyperspectral data with a line scanner. Today, the problem no longer exists due to general technical developments in the RPAS industry. This is a fast-growing industry with numerous new opportunities. New materials and lighter and more efficient batteries create better tradeoffs between the RPAS and its flight range, maximum altitude, and maximum payload [63]. The requirements for any platform used for hyperspectral survey for the purposes of vegetation stress inside and outside mine-contaminated areas are: flight velocity—as low as possible;flight altitudes—as low as possible (depending on the size of the observed object);swing and vibration—minimal for obtaining correct geometric images;controllability of platforms and navigation during flight, or GPS tracking during flight (flying the given routes and controlling the coverage area with images).

## 3. Methods

The main steps within the module for data acquisition (Figure 4) are described in the following sections.

### 3.1. Mission Planning and Image Acquisition

The use of airborne multi-sensor systems for remote-sensing applications have been increasingly spreading out nowadays. This is due to the flexibility and the ability to gather high-resolution imagery data and wide pallets of RPAS in the field of remote sensing [64,65,66]. The mission planning in airborne photogrammetry and remote-sensing applications depends on the system of acquisition and the selected platform. A detailed planning of a flight mission is a fundamental precondition for a successful acquisition of airborne data sets. It is important to emphasize that manned and unmanned surveys have differences in several aspects, such as flight duration, ground coverage and data capture techniques and the fact that the user does not have the direct control of the sensor (with RPAS), but they share a common background. The mission planning accounts for multiple steps that could be grouped as follows [46]: Selection of a suitable sensor and platform;Flight plan design; andAnalysis of the factors to be controlled during flight operations.

In order to scrupulously describe the mission planning in remote sensing, it is necessary to analyze the passive sensors according to their data acquisition geometry, which is generally based on the central perspective collinearity [67]. The image projection principle (ideal imaging process of a real object onto the image plane) is based on a geometric principle of the central perspective. The footprint of the image frame on the ground is closely connected to the relative altitude and the field of view (FoV). Based on the parameters of the camera (image resolution, pixel size, focal length) and flight height, the theoretical values of the GSD can be calculated, according to [68] as: (1)$$M_{b}=\frac{1}{m_{b}}=\frac{f}{h}$$
where *M_b_* is the scale of the image, *m_b_* the scale denominator, *f* the focal length and h the flight height, or [68]:(2)$$\frac{s}{f}=\frac{S}{h}$$
where *s* is the sensor size and *S* the size of the scanned scene, and [69]:(3)$$GSD=pel*m_{b}$$
where *pel* is the pixel size. With the known *GSD* and number of pixels in the sensor, the swat can be calculated. This is the value used for mission planning. The actual value of *GSD* is calculated after data acquisition, using calibration markers with striped black and white bars of decreasing width. This value depends mainly on the number of pixels, platform flight height, focal length and field of view of optic sensors.

Planning the flight route was complicated by the fact that the suspected hazardous areas are not one compact area, but fragmented over a large area, so the acquired data covered a much larger area than the area of interest (105 sq km). Single or parallel flight routes were planned and performed at different altitudes, repeated over certain areas. The standard flight altitude selected was 600 m above the mean terrain altitude, as these parameters proved the best and most economic for planning the flight route and the time required to take the images in each pass. At that altitude, GSD for the visible digital cameras was ≈7 cm, multi-spectral VNIR sensors ≈17 cm, for the panchromatic LWIR sensor ≈45 cm, and for ImSpector V9 ≈16 cm. As part of the effective flight time spent in the air, supplementary flights were carried out to obtain better GSD (lower-altitude flights), that is, a better insight into the terrain, where necessary. GPS and IMU data were also collected during each flight. These data provide the ability to create hyperspectral cubes and conduct georeferencing of collected images.

It should be emphasized that this module is not intended for photogrammetric surveys or planning, and the image collection is not always conducted so as to satisfy the strictest geometric conditions for their use (ensuring longitudinal overlaps greater than 60%, or transversal overlaps greater than 20%) [67,70]. However, it is still possible to mosaic the images and geocode the mosaics. The technical characteristics of the current sensors in the module allow enough overlap for photogrammetric surveys (particularly with a flight height of more than 500 m), but this is not a priority in conducting aerial non-technical surveys for humanitarian mine action purposes.

### 3.2. Image Quality Assessment

The interpretability of images is determined objectively and subjectively. A subjective assessment of the usability of images is done by an experienced scene interpreter, by visually reviewing the images with the use of contextual data. Therefore, a robust method is needed to allow an objective evaluation of the image quality which will correlate well with a visual, subjective judgement. For the objective determination of the interpretability of images, the image quality measure (IQM) [71] was used, with the Johnson criteria [72]. The IQM method is based on an analysis of image spectral density. Based on IQM values, National Image Interpretability Rating Scales (NIIRS) values are calculated, which provide the measure of interpretability, or usability. The image quality and interpretability are described by creating a NIIRS [73] scale (based on an analysis of the interpreter’s results), ranging from 1 (an image with the lowest interpretative quality) to 9 (an image with the highest interpretative quality). This has been used for over 20 years in the aerial imaging community (it was developed for military purposes) and was later adapted for civilian needs [74]. The concept underlying the NIIRS is that imagery analysts should be able to perform more demanding interpretation tasks as the quality or interpretability of the imagery increases. The method is not standard or developed for all types of objects for civilian use but gives a widely accepted assessment of the usability of images. The Johnson criteria, initially formulated as a method of predicting the probability of target discrimination, were created in 1958 [72]. The model uses the synergy of knowing the origin of how the image was created in the sensor and the interpreter’s experience and has been analyzed in detail in [75]. Johnson characterized the probability of detecting an object based on its actual resolution in the image. The concept has been substantiated by its own findings. He found that as the number of resolvable cycles across a target increased, so did the probability of an observer successfully locate a target. The Johnson criteria are the number of line pairs across a target needed for a group of observers to have a 50% possibility of discriminating the (target) object. From 1958 up to the present, this prediction and metrics model has been improved, although there is still no model that accurately predicts target detection in all inclement weather situations [75]. However, the method was used in AIDSS only to assess interpretability before visually interpreting the images, done by human interpreters, who make final decisions on target discrimination. That is, they determine the flight height in order to ensure the necessary size of GSD for the detection, recognition or identification of objects of interest with certain sensors by human interpreters. The methodology for extracting indicators of mine presence from hyperspectral VNIR images is described in [14,53]. 

Measuring positional errors in geocoded and/or georeferenced images or mosaics is difficult in these cases. This is because the places where data are stored are inaccessible (suspected hazardous areas or oil slicks at sea), and some scenes have changed drastically, so it is difficult to find the same points in the terrain before and after a crisis situation (for example, the munitions depot in Padjene before and after the fire, explosion and clearance). However, even in such situations, it proved possible to find a certain number of control points on the basis of which accuracy estimates were conducted for the operations implemented. For the orthomosaic, the positional error of these points was measured. The root mean square error (RMSE) was computed between check point coordinates determined on digital orthophoto and coordinates retrieved from georeferenced image mosaics, to assess the overall spatial accuracy of each dataset.

Before using and processing hyperspectral VNIR imagery, rigorous pre-processing steps were undertaken to ensure the quality, accuracy and interoperability of the data used. After parametric geocoding of hyperspectral data acquired using a V9 ImSpector line scanner, sensor radiance performance was inspected and validated. We used a calibration procedure based on the supervised vicarious calibration method [76] which included: (I) quality assurance of radiometric information, (II) stability and general performance analysis, (III) radiometric calibration and (IV) atmospheric calibration.

In the quality assurance procedure, MODTRAN was used to reconstruct the atmosphere above selected targets with ground-truth measured reflectance, and then compare the results with the at-sensor radiance obtained [77]. Two indices, Rad/Ref (at-sensor radiance divided by ground truth reflectance) and RRDF (radiance to reflectance difference factor) indices were used to spot faulty performance of the sensor prior to the next data processing stages.

In the present study, reflectance-based vicarious calibration was used for radiometric recalibration, as the image at-sensor radiometric data were calibrated by comparison against the modelled at-sensor radiance based on the in situ measured reflectance of the selected, well-defined ground targets. The simplest, fastest method for atmospheric correction is the empirical line method (ELM). It uses a set of ground targets of known reflectance to derive a relationship between sensor-spectral radiance and scene-spectral reflectance. ELM assumes that the radiance image contains some pixels with a known reflectance spectrum, and also that the radiance and reflectance values for each wavelength of the sensor are linearly related. Therefore, the image can be converted to reflectance by applying a simple gain and offset derived from the known pixels. 

The at-sensor measured radiance is given in an equation for each wavelength:(4)$$L_{s}=\;\frac{\tau \rho E_{0}}{\pi }+\;L_{path}$$
where E_0_ is the sun’s radiance above the atmosphere at a certain zenith angle, τ is the atmospheric transmittance, ρ is the surface reflectance and L_path_ is the selective scattering (Rayleigh and Mie) contribution to the sensor output [76]. Assuming that during the operation, the sensor keeps the calibration coefficients that were generated in the laboratory during the system calibration stage, Equation (4) is valid as it stands. In the case of a non-calibrated sensor (or divergence from the laboratory calibration), the achieved at-sensor radiance (L_s_) is a product of the real radiance multiplied by gain and offset coefficients that adjust the mis-calibrated laboratory information to at-sensor radiance as follows:(5)$$L_{s}=\left[L_{\left(gain\right)}\left(\frac{\tau \rho E_{0}}{\pi }+\;L_{\left(path\right)}\right)\right]+L_{\left(offset\right)}$$
where L_(offset)_ is the unknown noise that has entered the sensor since the time of the last laboratory calibration and L_(gain)_ is an amplification factor that depends on the sensor’s functionality and surrounding conditions that diverges from the laboratory calibration process [76].

### 3.3. Production of the Hyperspectral Cube

The robust structure of ImSpector V9 suits both industrial and scientific applications that require rapid, precise spectral measurements at low cost. The last configuration in AIDSS uses a narrow slit (8.8 mm × 50 μm) at the front end of the optical system and enables a spectral resolution of 4.4 nm of 80 channels in a spectral range from 430 nm to 900 nm. At the nadir, the system provides mapping of a narrow strip (0.333 × H) × (0.0028 × H), where H denotes the height above ground. The scanner was used to acquire reflectivity samples from the suspected hazardous area in several different types of terrain. The usefulness of the radiance is limited, due to its strong dependence on illumination, which can change during the acquisition mission. Thus, we did not attempt to measure the radiance, but calculated the reflectance coefficient. The reflectance coefficient is the ratio of the volume of electromagnetic waves recorded by the sensor to the volume of electromagnetic waves recorded by an aerial near the sensor. It is a property of the observed material and is equivalent under different illumination conditions. The spatial accuracy of airborne discrete measurements depends on the platform movements, positioning accuracy, and orientation system. When the system is placed on a mobile platform, it is possible to scan the terrain linearly, from the interval of line Δs. The interval of line Δs depends on the flight speed and frequency of storing fs images in the acquisition system (Figure 5a). Using the hyperspectral VNIR sensor and parameters, recording is manipulated via RECORDER program. For example, with a flight height of 750 m, vertical binning ×1 (for storage, the entire surface area of the PixelFly sensor is used), w = 218 m, GSD = 0.19 m. With a flight speed of 120 km/h (33.3 m/s), the frequency of image storage is f_i_ = 20, and the interval between the lines is 1.6 m.

To use the line scanner in full imaging mode (acquiring contiguous scan lines), it is necessary to find the optimal ground speed of the platform. It is a function of the required GSD and scanner imaging frequency, according to a simple Equation (6) used to arrive at optimal distance per second [78]:(6)$$GS=\frac{GSD}{f_{i}}$$
where:
*GS* = ground speed of the platform in (m/s); *GSD* = ground sampling distance in (m); and *f_i_* = Imaging scan period in (s).

The maximum frame per second is around 24 Hz and depends on radiometric parameters during collection reflected radiation (exposure time and sensor sensitivity). Line scanning of the ImSpector V9 hyperspectral VNIR sensor demands a very complex calibration procedure and time-consuming processing (Figure 5a). Software solutions in the Matlab package have been developed to produce a raw hyperspectral cube (Appendix A), which is then parametrically geocoded using the PARGE 2.3 software package [79].

### 3.4. Geotagging and Triage

Based on GPS data, geotagging [80] and parametric geocoding [81] can be performed (Figure 6). The mine scene interpreter can then conduct triage on the geotagged raw images. It is important to emphasize that raw images are interpreted, because using orthography procedures on hilly, mountainous terrain (where there are great differences in elevation) can lead to geometric deformities on the images, making high-quality interpretation impossible. In addition, triage is carried out by inspecting raw images and selecting those where indicators of mine presence have been detected (Figure 6b). If indicators of mine presence are detected on several neighboring images, mosaicking is performed. After that, the selected images and mosaics must be geocoded in order to locate the indicators of mine presence in space, and so that images from different sensors can be co-registered.

### 3.5. Mosaicking and Georeferencing

Image mosaics can be produced and geocoded manually and automatically. Geocoding of image mosaics has been performed using automatically or parametrically specialized PhotoScan (AgiSoft Metashape, Russia) software for visible, multispectral or thermal imagery or ENVI (Harris Geospatial Solutions, Inc., Boulder, CO, USA) for hyperspectral cubes. Digital orthophoto maps are the best reference for manual geocoding of images or mosaics of images. Their main application is in photo echometrics of various indicators in a suspected hazardous area, where ground control is neither available nor needed, and where directly georeferenced digital imagery is acquired to solve the exterior orientation problem [21]. However, if necessary, georeferencing can be performed without in situ ground control points [82]. 

Digital image mosaics and digital surface models (DSM) are produced using a feature-matching algorithm of structure-from-motion (SfM, [83]) that analyses all images of the aerial data set and searches for matching points. PhotoScan Professional (Agisoft, Russia) uses this concept and was selected for geocoding the images and image mosaics of matrix cameras. While working with it, PhotoScan proved to be robust software and less demanding in terms of entering parameters for a photographic survey. All the subsequent procedures for producing the image mosaic were conducted without significant difficulties. To improve the absolute spatial accuracy of the image mosaics, ground control points were manually distributed within the imagery. In this system, the information from GPS and IMU was used to perform aerial triangulation without classic ground control points marked on the ground. Small details were selected for ground control points, which were visible on the digital orthophoto map and raw images. PhotoScan Professional provided an interface to mark the location of a ground control point on each image and its location was then automatically marked on all the images where this ground control point appeared.

The procedure for parametric geocoding means assigning precise spatial positions for each pixel in the hyperspectral cube. This requires knowing in advance the elements of external orientation (GPS and IMU data) for each line of the hyperspectral cube. The parametric process begins with an estimate of the theoretical view vector ($\vec{L}$) which is the imaginary line of sight to the current pixel, oriented from a horizontal platform facing north. This vector must be turned in three dimensions to obtain the effective view vector ($\vec{L_{t}}$) [81]:(7)$$\vec{L_{t}}=\;Κ*\Omega *\Phi *\vec{L}$$
where Κ, Ω and Φ are the coordinate transformation matrices for roll, pitch and true heading, respectively.

Data preparation for parametric geocoding includes adapting the format of the data received by the system. Before arranging spectral lines in a raw hyperspectral cube, mean data are created on insolation (data written using the aerial), and the coefficients of reflectance and subtracting the dark current are calculated. The data are finally converted to create a visual pyramids in BSQ format with ENVI heading, which allows direct reading of the raw hyperspectral cube in the PARGE program for parametric geocoding. These operations are carried out automatically, with code written in the Matlab development environment specifically for this purpose. Next, data is synchronized with iMAR (inertial measurement unit) and recording so that the corresponding orientation elements for each image are found using interpolation of linear and angle orientation elements (GPS and IMU data), and correspond to the moment when each image was taken (Figure 7). This process is also fully automated with code written in the C # development environment. This means that only data recorded during the system operation, using its frequency, are recorded, while the rest of the big data from iMAR is discarded as superfluous and does not overload the computer resources. The program can be used, in addition to interpolation, to acquire data in a format adapted to PARGE parametric geocoding. Because of the mutual independence of data groups (images, metadata on images, and data from GPS and IMU), and the different programs used to access them, this kind of configuration allows parallel processing on multi-core systems, which speeds up overall data processing considerably. 

## 4. Results

### 4.1. Sensor Pods and Platforms

Sensor pods were designed and made especially for each individual platform. For the Mi-8 and Bell-206 helicopters (Figure 8a), two types of large aerodynamic sensor pods were made (weighing over 20 kg). The Gazela helicopter has a hole in the floor, which made the development of the sensor system easier (Figure 8c). For the RPAS platform, smaller sensor pods were designed and made, depending on their payloads (the weight of the system varied from 3.5 to 5 kg, Figure 8c). Micro computers were built into these sensor pods so that the sensors could be activated remotely from the ground. The results of the operational testing performed for platforms: Mi-8 and Bell-206 helicopters, RPAS s X8 MR and 8 ZERO and blimp are shown in Table 2.

### 4.2. Analysis Vibration of the Sensor Pod Installed on the Mi-8 Helicopter

Experience with different aerial platforms (Bell-206, Mi-8, and Gazela helicopters, RPAS Fenix fixed-wing, and several multirotor RPAS s) used for airborne imagery acquisition has shown that sensor vibration is the main cause of image blurring. There are no published data about the vibration of these aerial platforms. This fact motivated us to analyze the vibration which occurred in a pod containing electro optical sensors (Figure 9b,c). We selected the pod installed underneath the Mi-8 helicopter. Data on vibrations were collected using the inertial positioning system (iMAR), along with data provided by the GPS receiver. By analyzing the collected acceleration in flight direction X, in a left or right direction Y, and in a vertical direction Z (Figure 9b), we were able to identify several flight phases: (a) engines on, helicopter on the ground, (b) take-off, helicopter climbing, and (c) the stationary phase of the helicopter’s flight. Since only the stationary parts of the flight route were usable for image acquisition, the spectral analysis of vibration was carried out on segments of this phase (Figure 10a,b). The results obtained (Figure 10c–e) enabled the design, development and implementation of passive vibration damping to decrease image blurring.

The analysis of spectral density of the power of vibration showed that vibration occurred at the following frequencies: ~3.2 Hz, ~4.0 Hz and ~7.5 Hz. The first is the direct frequency caused by the rotation of the main rotor (without blades). The second is the frequency caused by aliasing vibration at 16 Hz, which is the result of the rotation at 3.2 Hz of the five blades of the main rotor depicted in reverse under 10 Hz; 16 Hz − 10 Hz = 6 Hz, which is shown counting down from 10 Hz and gives ~4 Hz. The last is the actual frequency of vibration or occurs due to aliasing wrongly depicted frequencies ~12.5 Hz, for unknown reasons. Based on these measures and the results of the analysis, passive dampers were designed for 3.2 Hz, 16 Hz, 7.5 Hz and 12.5 Hz using Enidine vibration insulators (Figure 9a).

During the 2010 AIDSS mission in Bosnia and Herzegovina, the first tests were conducted (to reduce the colour blurring in images after passive vibration damping), in which Enidine vibration isolators were (a) blocked (no vibration damping), or (b) isolators were enabled to damp the vibrations. The images were collected in both cases and excellent results achieved: images with activated isolators could be zoomed seven to eight times before discernible blurring occurred and have larger standard deviation (27.921 without isolators v.s 30.906 with isolators, Figure 11). When vibration isolators were disabled, zooming two or three times made the images blurred.

### 4.3. ImSpector V9 Line Scanner Calibration Procedure Results

A quality assurance procedure performed via MODTRAN reconstructed the atmosphere above for selected targets with ground-truth measured reflectance. The calculated at-sensor radiance and modelled at-sensor radiance based on in situ measured reflectance. The results obtained demonstrated obvious radiometric/spectral defects, which had been corrected before undertaking any other action. First, there was full saturation of the brightest targets, which is a typical sign of the poorly calibrated or uncontrolled dynamic range of the system. Second, a highly distorted albedo sequence and lack of gas absorption were clearly demonstrated. Therefore, the spectral/spatial saturation and local spectral stagnation were inspected together with the spectral accuracy of atmospheric gas absorption of the at-sensor radiance data. Figure 12c shows image at-sensor radiance against modelled radiance based on in situ reflectance measuring the reflectance of one selected target. The comparison reveals clear spectral/radiometric differences and distortions. Calculated Rad/Ref and RRDF indices indicated the faulty performance of the sensor. 

In order to avoid saturation of the brightest targets on the image at-sensor radiance, we made an additional RRDF calculation, which did not involve the brightest spectra (radiance and reflectance). Since the sensor was not performing well, the estimation of L_(gain)_ and L_(offset)_ was performed by vicarious calibration. Reflectance-based vicarious calibration is used for radiometric recalibration, as image at-sensor radiometric data are calibrated by comparison with the modelled at-sensor radiance based on in situ measured reflectance of selected, well-defined ground targets. During the validation stage, two ground targets were selected and compared. It is important to note that these targets were never included in the recalibration process. They were spectrally measured in situ (reflectance) and stored for the validation stage. At this stage, we can declare that the recalibrated image at-sensor radiance obtained is ready for the next stage—atmospheric correction via calculated coefficients. The final results of the calibration procedure obtained in ENVI (Figure 12d,e).

### 4.4. Imagery

Images collected with visible digital cameras had the best GSD and depicted the largest recorded terrain, compared to other sensors (Figure 13). Hyperspectral VNIR sensor images have poorer GSD than the multi-spectral VNIR sensors (Figure 13a), but they exploit the full spectral dimension, which better reflects the continuous nature of actual spectra. Panchromatic LWIR Photon 320 sensor also have poorer GSD than the visible digital cameras (Figure 13b), however, they are used to detect indicators of mine presence via their thermal characteristics. LWIR images acquired by a panchromatic LWIR sensor showed trenches that could not be seen from the ground. These images revealed details not seen on multi-spectral VNIR images, which justified their use in the module (Figure 14a,b). On the multi-spectral VNIR image inside the suspected hazardous area the top part of the trench was only just showing, while on the LWIR image, it can be seen in its entirety. Detection of strong indicators of mine presence is carried out by mine scene interpreters through visual interpretation of images collected with visible digital cameras and LWIR sensors.

The confidence values of subjective assessment of identifying indicators of mine presence by mine scene interpreters, and an objective assessment of image quality using NIIRS are used (Table 3). The values of 5 to 7 on the NIIRS scale (Table 3) allow detection of fortification objects on images taken by selected sensors. The calculation of the percentages of detection, recognition and identification probability for a single strong indicators of mine presence according to the Johnson criterion was performed (Figure 15). The critical value of the trench (its normal width) was determined at 0.6 m, and for the bunker (its normal radius) at 2 m.

## 5. Discussion

AIDSS is the first mine-action technology to successfully combine remote sensing with advanced intelligence methodology for non-technical surveys proposed to mine action centres worldwide, because it is adaptable to specific terrains and situations. Therefore, the primary goal of the AIDSS module for aerial data acquisition in crisis situations is to acquire high-quality inputs for further processing according to the AIDSS methodology. The AIDSS multi-sensor imaging system includes multiple sensors integrated onto the same substrate forming a single multi-sensor platform but does not provide the known consistent physical relationship between the multiple sensors. However, it ensures synchronization of all sensors and computers guiding them, and so links the storage times of individual images with IMU and GPS data. Radar technology was not considered within the scope of AIDSS research, but we encountered aerial radar technology in the SMART [84], and TIRAMISU [85,86] projects. Synthetic aperture radar from SMART was used in 2001. It had four wavelengths, polarization modes, and showed the potential to distinguish between several kinds of target (trenches, pioneering vegetation invading former cultivated areas). Between 2001 and 2019, significant advancement in radar technology occurred which could be applied to mine action, even for the direct detection of land mines, for example [87]. However, the most important aspect was that AIDSS could be used in a non-technical survey, not for locating mines, but for strong indicators of mine presences, and it could be used in synergy with other technologies applied in humanitarian mine action. 

The efficient use of AIDSS and thus the module for aerial data acquisition, depend on a quality analytical assessment of existing data and the creation of a list of indicators, in cooperation with experts in particular crisis situations. As part of the module creation process, software solutions have been developed in the Matlab environment for making a hyperspectral cube of sequential spectral lines collected by a hyperspectral VNIR system consisting of an ImSpector V9 line scanner and PixelFly matrix camera. In this way, the line scanner is can be used in full imaging mode. A calibration procedure for the hyperspectral data also has been established.

Mission planning for the purpose of aerial data acquisition regarding suspected hazardous areas is specific and is not intended for classical photogrammetric surveys. Suspected hazardous areas are mainly fragmented over a large area, so flight routes need to be planned accordingly. Within mission planning of aerial data acquisition, Johnson criteria are used to determine flight altitude, in order to ensure the necessary GSD size for detecting, recognizing or identifying an indicator. After a mission, image-quality assessments (IQM and NIIRS values) are conducted for the purpose of analyzing the interpretability and usability of the images. Thus, all procedures within the survey mission are optimized, and high-quality imagery is provided for further operations within AIDSS.

Visible digital cameras provide GSD from 1 to 11 cm for flight heights from 50 to 1000 m, multi-spectral VNIR and hyperspectral VNIR sensors from 2 to 28 cm, and thermal infrared sensors from 3 to 60 cm. For an initial insight into the scene, imaging from a height of 600 m was conducted All the images of the visible sensors had sufficient GDS to detect the target AIDSS indicators according to the NIIRS scale and Johnson’s criteria (Figure 15). Other sensors were used to supplement the insight into the situation at the scene (potential elimination of dubious elements when drawing conclusions) and for specific purposes. In the case of the hyperspectral VNIR sensor, to stress vegetation spectral response inside and outside the minefields and image classification of the oil slick. Based on GPS and IMU data, orthomosaicking, digital surface models and parametric geocoding can be performed within this module, although no classical photogrammetric survey was performed. A problem encountered in geocoding gathered images in this way was the impossibility of setting ground control points within a dangerous suspected hazardous area, or where entry was prohibited. The problems of small longitudinal and transversal overlaps on images, and the lack of ground control points, were resolved using a feature-matching SfM algorithm.

The specifics of this module are also reflected in the easy adaptation of sensor pods to various types of platforms in relation to the task to be performed. This kind of system is suitable for collecting images of large and/or relatively small areas, since it can be mounted on helicopters and RPASs. Data collection is economized by combining the use of both types of platform. All sensor pods were designed and produced for a specific platform. The module ensures the stability and reliability of aerial data acquisition on each platform via an independent power supply. Wire rope dampers for passive vibration damping were designed, developed and installed in sensor pods to reduce image blurring. The size of the fragmented parts influenced the decision on which platform, or platforms, would be selected for data collection. Multiple platforms can be used for one suspected hazardous area, depending on the size of the fragmented parts of it.

Operational testing performed for platforms: Mi-8 and Bell-206 helicopters, RPAS s X8 MR and 8 ZERO and blimp (Table 2). RPAS was used to record individual parts of scenes of particular importance, with sufficient GSD to identify the indicators in them. The blimp was rejected as a suitable platform due to several drawbacks: (I) sudden, large changes of roll and pitch values during flight caused major distortions of hyperspectral images, (II) it required relatively high velocity and altitude values to obtain stable platform flight regime, (III) there was low controllability during flight.

The results (Table 2) showed that the helicopters tested were the right choice for surveying larger regions of interest with coarser spatial resolution data requirements. In comparison with RPASs, helicopters have higher payload limits and endurance, but velocity and altitude values are much higher, as are vibrations. When choosing between the smaller helicopters (Bell-206, Gazela) and the larger Mi-8, the following should be taken into account: smaller helicopters have lower altitudes, velocity, and vibrations, but have payload limitations—they can carry a multispectral survey system and only one system operator. In addition, validation tests of the helicopters show that controlling the platform when executing planned survey routes can be a challenge, since correction of the yaw (direction angle) dramatically enlarges the values of the roll and pitch parameters (swinging). The results of the operational testing performed for two RPASs (Table 2) showed that these particular models struggled to maintain stability due to borderline payload, which caused deviation from planned routes. Payload limitation also dramatically affected the endurance of the system, restricting operational deployment. On the other hand, these platforms performed at low altitudes and velocities with low vibration and showed a satisfactory reduction in the value of pitch and roll parameters. For the operational deployment of the hyperspectral VNIR sensors, heavy lift RPAS should be used (RPAS 8 ZERO, Figure 3b).

Of course, there are limitations to the use of this module. The main limitation is aerial data acquisition over forested regions and snow-covered mountains, where it is impossible to see and detect indicators beneath the foliage. Furthermore, it is hard to find and set ground control points on the terrain for the purpose of more accurate image georeferencing. The problem is greater in evergreen forests, but it could be solved by introducing Lidar into the module. However, the use of program tools such as UgCS Mission Planning Software (https://www.ugcs.com/) within the AIDSS data acquisition module may reduce such limitations and enhance the value of the entire system. Another problem in such areas is the difficulty of accommodating to strong winds, which prevented data acquisition on the planned route. Air currents affected the stability of all platforms (especially for RPASs), requiring flight downwind or parallel to the wind. Therefore, the flight route in such cases depended strongly on atmospheric conditions. In sparsely populated mountain areas, a problem may also be created by the lack of satellites for orientation and navigation during aerial data collection. Furthermore, RPAS-based applications limitations are largely related to the currently large weight of the hyperspectral system and power supply requirements of its sensors, highlighting the need for future miniaturization in such devices.

The module architecture allows the integration of more sensors, replacing existing ones or choosing other platforms to improve the module’s results. For example, light detection and ranging (Lidar) is a potential sensor that could contribute to better understanding of the situation in the suspected hazardous area and discrimination indicators in relation to the environment. Recent advances in RPAS technology, combined with lightweight sensors and a power supply (battery) provide a greater autonomy and longer flight time for aerial data acquisition at high resolution. These opportunities should be exploited in the subsequent projects.

## 6. Conclusions

Within the framework of the research conducted in six international and domestic scientific projects, an AIDSS module for aerial data acquisition in crisis situations and environmental protection was developed, tested and implemented in operational (real) conditions. Designing, production and use of AIDSS module for aerial data acquisition are interdisciplinary tasks that require harmonious and efficient behaviour among components at play. This paper has described the role of that module within AIDSS. The module was found to be effective at aerial data acquisition in crisis situations and environmental protection, and especially in humanitarian mine action. It adds significant value for the demining community because it is designed to meet the real needs of mine-clearance experts and encompasses all the necessary actions from analytical preparation, data collection and pre-processing data for DSS. In spite of the fact that the module was not conceived as a system for strict photogrammetric recording of inaccessible terrains, it provides the potential for georeferencing images and creating orthomosaics. This demonstrates the usability of its results in further processing and obtaining final results of AIDSS or some similar decision-supported system about crisis situations and environmental protection. The specificity of this module is also that it is customizable for different types of platform, for which special pods are constructed with different sensor and electricity power supply configurations. Mostly off-the-shelf equipment and software were used, but some software solutions for the image collection of hyperspectral VNIR data and production of raw and parametrical georeferenced hyperspectral cubes were made especially for this module in Matlab programme. One type of programme developed and described in the research is the RECORDER for control and management (selecting various parameters for the best adjustment to atmospheric conditions) during recording by DuncanTech MS-3100 and ImSpector V9 + PixelFly VNIR sensors. The development of the system began using helicopters as platforms and continued using RPAS, as they developed and increased their flight and load-bearing characteristics. The module was developed to be as independent as possible of the platform used, with the potential to adjust and use it on various airborne platforms with minimal modifications.

## Figures and Tables

**Figure 1 sensors-20-01267-f001:**
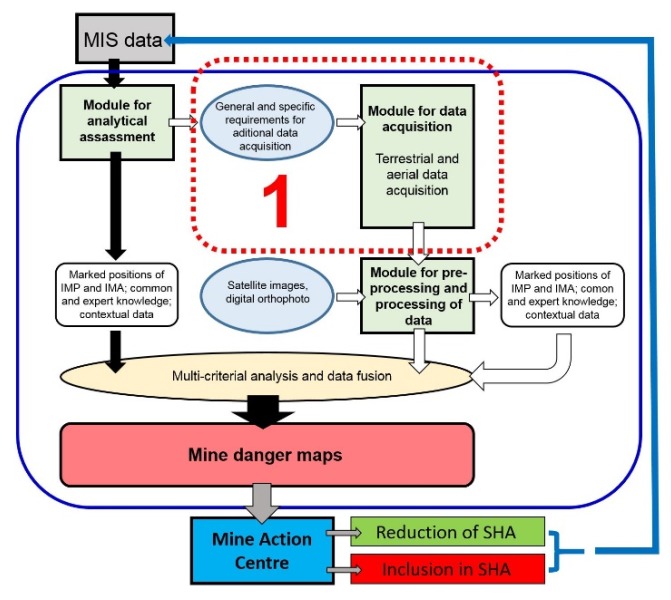
Scheme of Advanced Intelligence Decision Support System (AIDSS) methodology. 1—Module and requirements for data acquisition.

**Figure 2 sensors-20-01267-f002:**
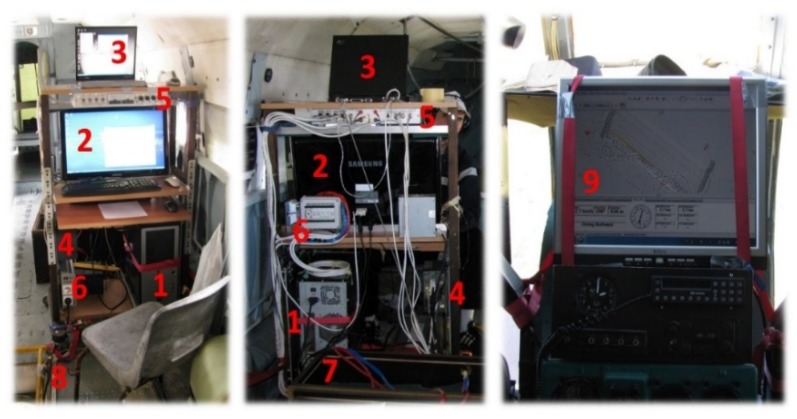
Control table with module control system component inside Mi-8 helicopter: (1) personal computer (PC) for manipulation with Nikon D90 and Photon 320; (2) monitor for PC and industrial controller (switching is performed as needed during recording); (3) laptop for manipulation with inertial measurement unit (IMU); (4) industrial controller for manipulation with MS4100 and ImSpector + PixelFly (not part of the first system configuration, built in 2009); (5) junction box of the electric power system; (6) converter from 28–30 V direct current (DC) to 220 V alternating current (AC) for PC and monitor; (7) large battery (210 Ah, 75 kg) for power supply; (8) cables for connection with sensor; (9) navigation monitor located in the cockpit.

**Figure 3 sensors-20-01267-f003:**
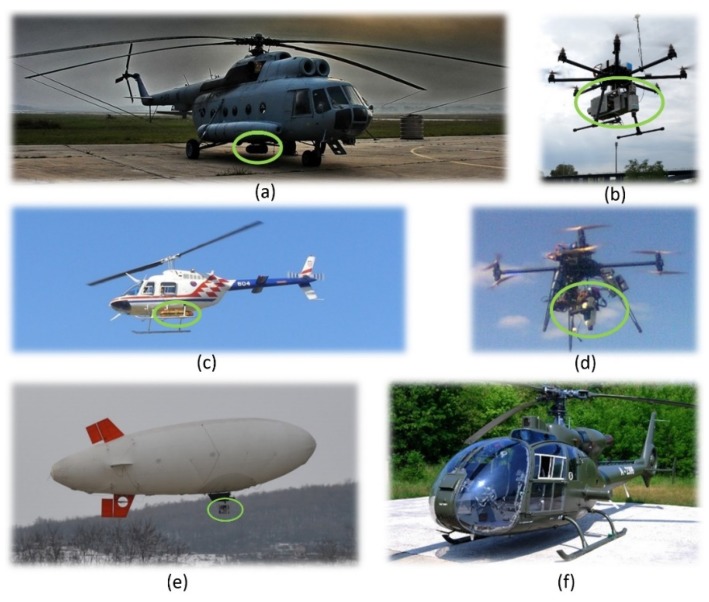
Platforms for aerial data acquisition system of AIDSS and sensor pods (in the green ellipses) on (**a**) Mi-8 helicopter, (**b**) remotely piloted aircraft system (RPAS) 8 ZERO, (**c**) Bell-206helicopter, (**d**) RPAS X8 MK, (**e**) blimp, and (**f**) Gazela helicopter.

**Figure 4 sensors-20-01267-f004:**
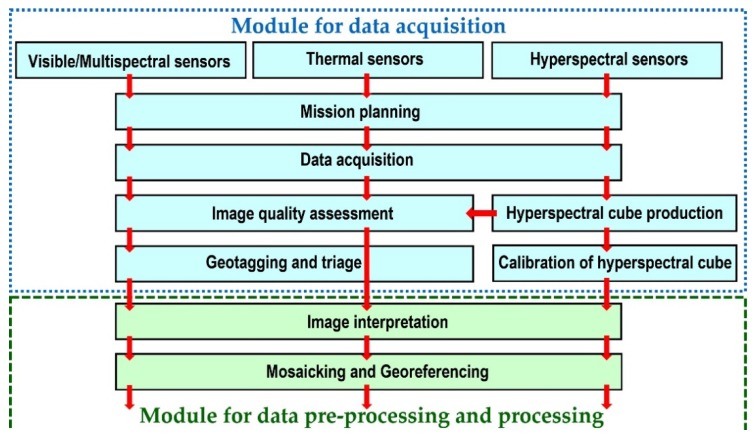
The main steps within module for data acquisition.

**Figure 5 sensors-20-01267-f005:**
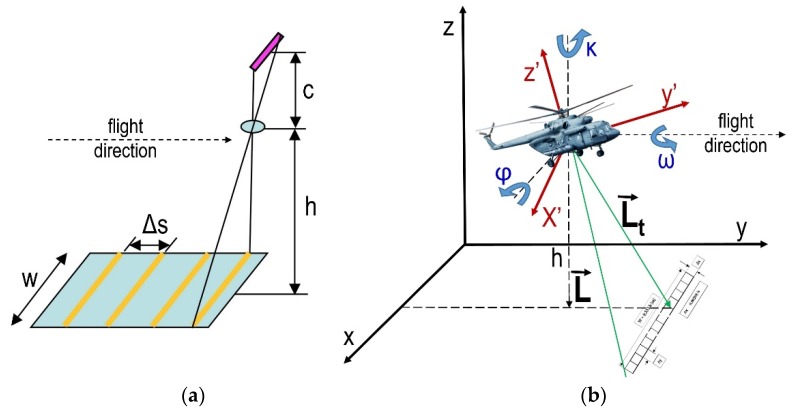
(**a**) Geometry of transference of the ImSpector V9 and PixelFly hyperspectral visible, near infrared (VNIR) systems, (**b**) transformation of theoretical view vector $\vec{L}$ to effective view vector $\vec{L_{t}}$. κ, ω and φ denote roll, pitch and true heading angles respectively.

**Figure 6 sensors-20-01267-f006:**
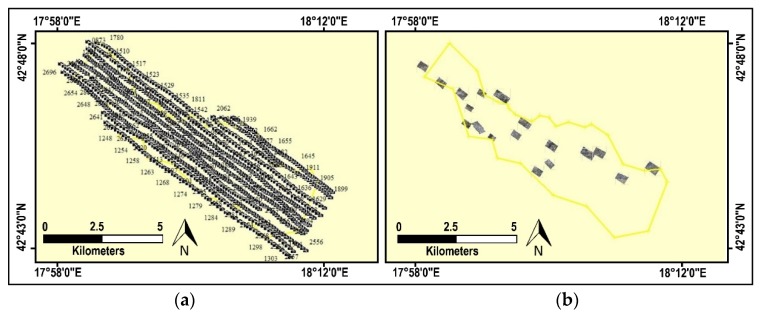
Examples of depictions of (**a**) geotagged images and (**b**) selected and georeferenced images, after triage, which contain indicators of mine presence in one suspected hazardous area (yellow polygon).

**Figure 7 sensors-20-01267-f007:**
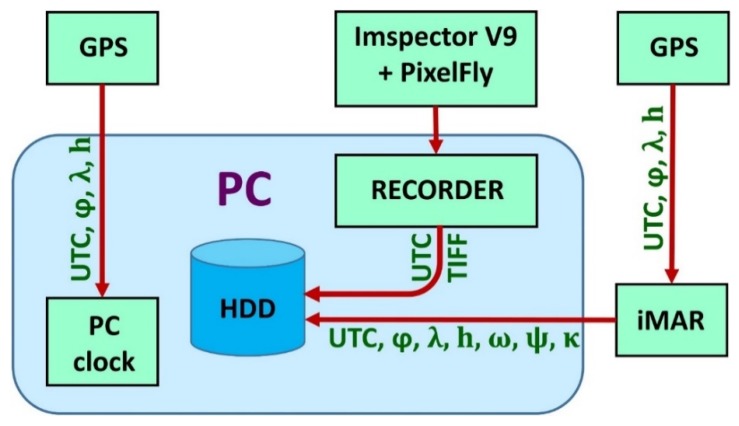
Schematic view of the integrated hyperspectral imaging system comprising ImSpector V9 + PixelFly camera.

**Figure 8 sensors-20-01267-f008:**
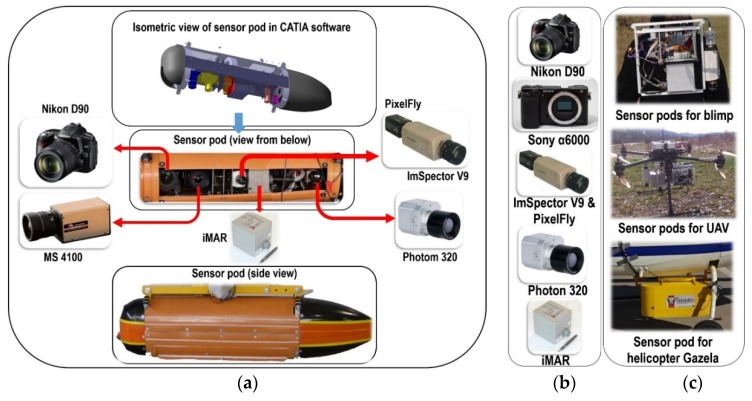
(**a**) Example of the first sensor pod designed and made with associated sensors for the Mi-8 and Bell-206 platforms. (**b**) Sensor used and (**c**) examples of a sensor pod designed and made for blimp (top), unmanned aerial vehicle (UAV, in the middle) and helicopter Gazela (bottom).

**Figure 9 sensors-20-01267-f009:**
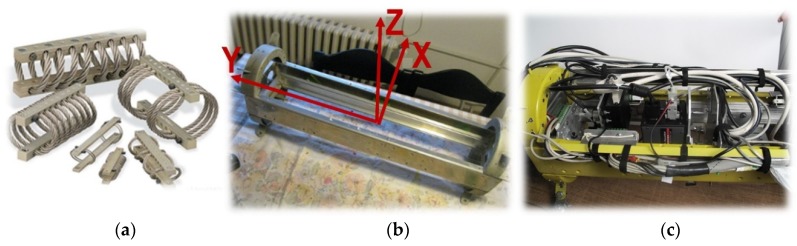
(**a**) Wire rope vibration dampers (ITT, 2019), (**b**) coordinate axes in sensor pod, (**c**) wire rope dampers installed in the sensor pod.

**Figure 10 sensors-20-01267-f010:**
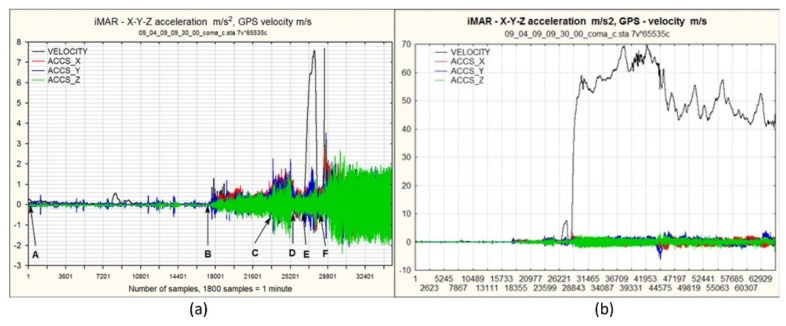
(**a**) Vibration phases: AB engines off, BCD engines on, EF increase power and take off, > F flight; (**b**) the diagram shows the speed (velocity, m/s) and acceleration dependence on the *X*, *Y* and *Z* axes (ACCS_X, ACCS_Y, ACCS_Z, m/s²) in relation to the number of samples. Speed was calculated based on data from the GPS receiver with a frequency of 1 Hz, and acceleration was read from iMAR with a frequency of 20 Hz on the measured platform accelerations per axis: (**c**) *X* axis, (**d**) *Y* axis and (**e**) *Z* axis.

**Figure 11 sensors-20-01267-f011:**
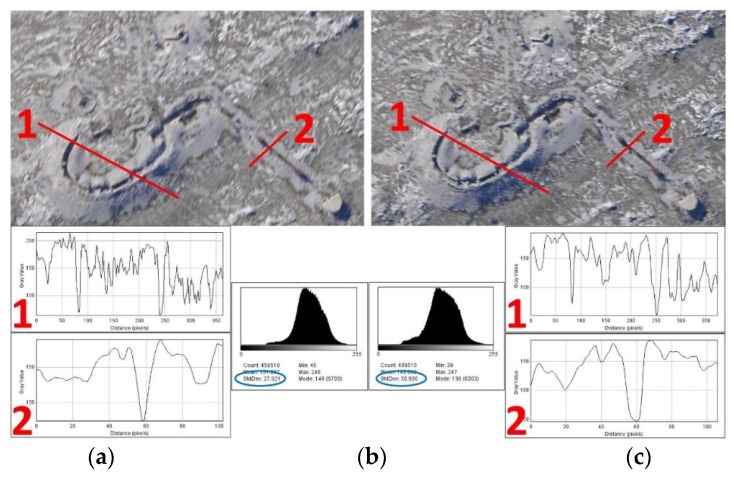
(**a**) Images (M_b_ ≈ 1:12,000) collected without damped vibrations with two profiles over the indicators of mine presence (trenches). (**b**) Passive damping of vibrations keeps the image sharper with two profiles over the indicators of mine presence (trenches). (**c**) Histograms of images with standard deviations (in blue ellipses).

**Figure 12 sensors-20-01267-f012:**
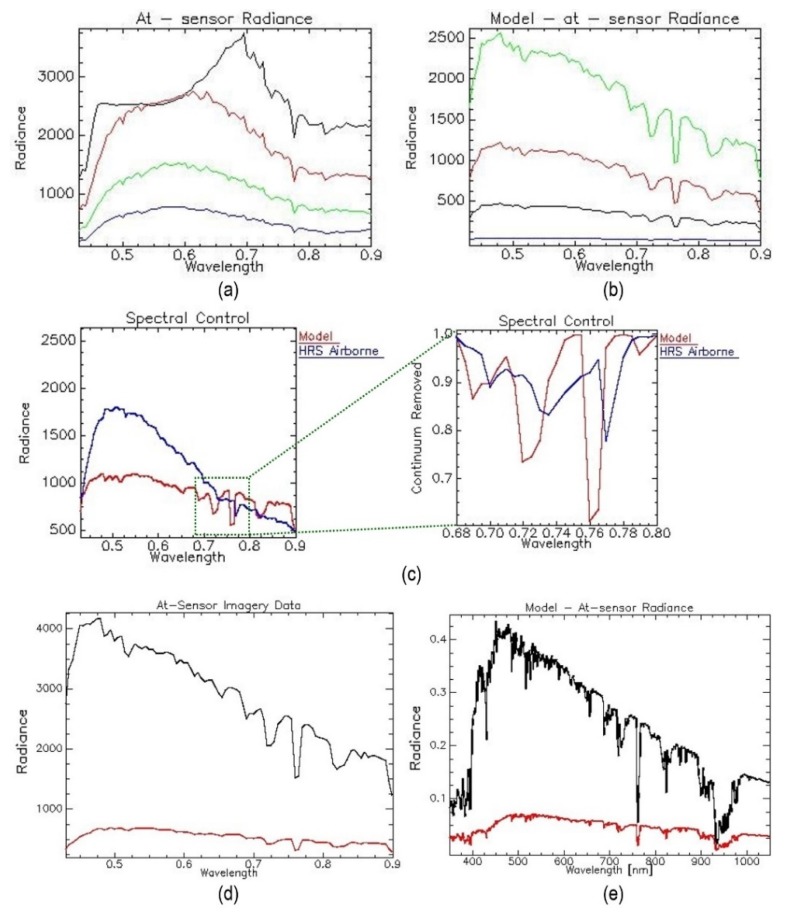
(**a**) Image at-sensor radiance and (**b**) modelled at-sensor radiance based on in situ measured reflectance (right side) of four selected ground targets. (**c**) Illustration of radiometric/spectral defects - investigation of image spectral accuracy based on atmospheric gas absorption (blue spectrum) compared with simulated radiance (red) spectrum for the same ground-truth target. (**d**) The modelled at-sensor radiance based on in situ measured reflectance and (**e**) image at-sensor radiance.

**Figure 13 sensors-20-01267-f013:**
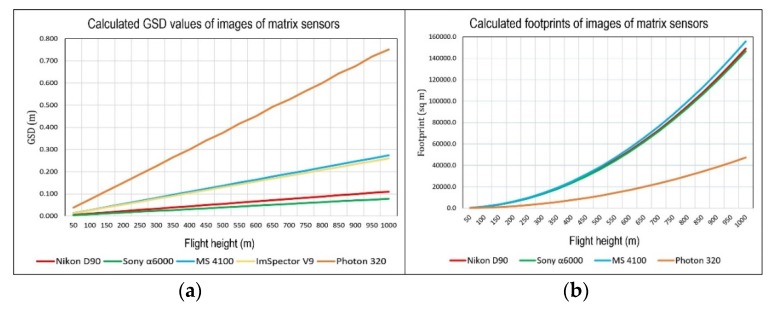
(**a**) Theoretical values of ground sample distance (GSD) of images of more significant AIDSS sensors for flight heights of 50 to 1000 m; (**b**) Theoretical values of image footprints of matrix sensors.

**Figure 14 sensors-20-01267-f014:**
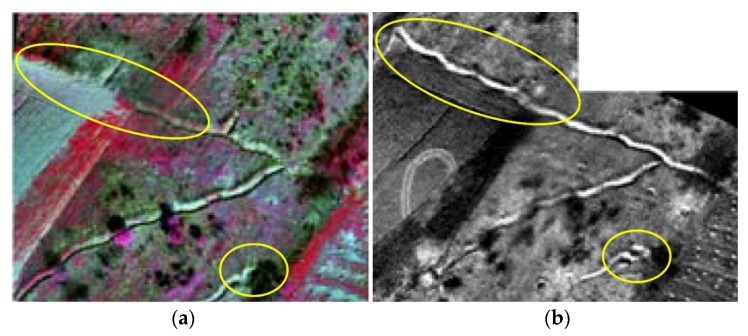
Examples of images of part of suspected hazardous area (from a height of 260 m, M_b_ ≈ 1:10,800) collected using sensors (**a**) multi-spectral VNIR and (**b**) panchromatic LWIR on which trenches can be seen (yellow ellipses).

**Figure 15 sensors-20-01267-f015:**
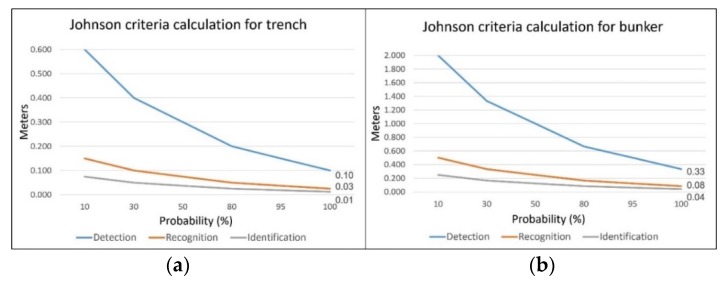
GDS size for the detection, recognition and identification of (**a**) a trench and (**b**) a bunker. Both diagrams show the GSD values for 100% detection, recognition and identification of the objects on the image.

**Table 1 sensors-20-01267-t001:** The main characteristics of image sensors, number of spectral bands and spectral ranges used within the module for aerial data acquisition.

Camera	Sensor Type	Sensor Size (mm)	Max Image Size (pixel)	Radiometric Resolution (bit)	Spectral Bands	Spectral Range (µm)
Nikon D90 DX	CMOS	23.6 × 16.8	4288 × 2848	12	3	0.4–0.7
Sony α6000	APS-C	23.5 × 15.6	6000 × 4000	12	3	0.4–0.7
DuncanTech MS3100	3 × CCD	7.6 × 6.2	1392 × 1039	8 and 10	4	0.4–1.0
Redlake MS410	3 × CCD	14.2 × 8	1920 × 1080	12	4	0.4–1.0
Thermovision 1000 FLIR	mini-STIRLING cooled		600 × 400	8	1	8–12
FLIR Photon 320	uncooled Vanadium Oxide microbolometer		324 × 256	14	1	8–12
Imspector V9 + PCO PixelFly	line scanner, CCD	8.6 × 6.9	1280 × 1024	12	up to 95	0.4–0.93
CUBERT UHD 185	Si CCD		1000 × 1000 50 × 50	12	125	0.45–0.95

**Table 2 sensors-20-01267-t002:** Summarized results of tests performed in different operational conditions for the following platforms: Mi-8 and Bell-206 helicopters, RPAS X8 MR, RPAS 8 ZERO, and a blimp with hyperspectral line scanner V9, multirotor UAV with hyperspectral frame sensor UHD-185.

Platform	Min. Velocity (m/s)	Swinging	Payload (≈4 kg)	Vibration	Controllability
Mi-8 helicopter	33	Relatively stable	High	High	Good
Bell-206 helicopter	20	Relatively stable	Relatively high	Medium	Good
RPAS X8 MR (smaller)	1–4	Medium—roll span: 2.8°–4.5°,	Borderline	Low	Executed routes significantly deviated from the planned ones
		—pitch span: 1.2°–2.0°			
RPAS 8 ZERO (smaller)	4–5	Medium—roll span: 3.3°–5.6°,	Sufficient	Low	Executed routes significantly deviated from the planned ones
		—pitch span: 1.2°–2.0°			
Blimp	4–5	Very high—roll span >20°,	Sufficient	Low	Difficult to navigate during flight
		—pitch span >10°			(large yaw)

**Table 3 sensors-20-01267-t003:** Some examples of radiometric characteristics, theoretical GSD of selected images with indicators of mine presence, cameras with which the images were collected and the subjective confidence of the human image interpreter in his findings.

Indicator of Mine Presence	Sensor (Image No.)	GRD (m)	IQM	NIRS	Subjective Confidence of the Human Image Interpreter
Bunker	MS-4100 (313) Nikon D90 (338)	0.29 0.10	0.00893 0.00107	5.38 5.51	1
Drywall					1
Trench					1
Trench	Canon 5D (1310)	0.19	0.00913	5.28	1
Battlement					1
Shelter					1
Unexploded ordnance (UXO)	Sony α6000 (1308)	0.03	0.0595	6.7	1

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
