# Peer review of "The AIDSS Module for Data Acquisition in Crisis Situations and Environmental Protection"

_sensors, 2020, doi:10.3390/s20051267_

Round 1
Reviewer 1 Report
The manuscript describes a module for mapping areas of fortification objects (e.g., exploded munitions depots) as well as land mines to develop hazard maps. This manuscript is suitable for publication after revisions. The methods section needs to be reduced and streamlined. There is much extraneous material that is either not needed or could be moved to a supplemental section. One example of this is the SfM section that describes how the metashape software works, or how the geocoding works, which doesn’t appear necessary. Section 2.1 specifically needs to be laid out in a more logical fashion. The section jumps around between the need for the module, how it was developed, and its sections. I recommend the authors start from an outline and re-arrange the section and sentences into a more logical fashion.
The results section is also largely methods, and includes comparison and analysis of the mapping methods that should have be laid out in the methods section. The results section should be laid out to focus on the results of the methods laid out in the methods section.
Lastly, the discussion section is of 1.5 pages in length, and in comparison with a manuscript of 31 pages, does not appear adequate. The other portions of the paper should be more in line with this length, with perhaps more detailed discussion of the sensor packages relegated to a supplemental section.
There are also too many acronyms in this paper, and especially in its abstract, and include acronyms that are both unnecessary and not commonly used in remote sensing. Please reduce use of acronyms to a core number. It is not possible to keep the number of acronyms in working memory while reading this paper. The citation format is a little odd in that when multiple references are referred to, they are not set off in the same brackets – but scattered around sentences, such as in e.g., Line 106, 119, 125-6, 231-2, etc. These references should be consolidated into one set of brackets.
Additional line-based suggestions follow.
Line 21-22 – “Display the scene in various ways” is vague and unspecific.
Line 23-28 – This sentence is not quite grammatical right now, I recommend revising to, “ and some software solutions were also developed.” I do think this sentence could be improved – having all the material in parentheses is not very clear
Line 35: revise “(about 10.5% (Human Rights Watch, 1999))” to “(about 10.5%; Human Rights Watch, 1999)
Line 63: sensors
Line 64: numbers less than 10 should be spelled out
Line 73, what the “Impact of Flooding on Mine Action” is, is unclear. Is this a government program? Or a module? If a module why is it capitalized and the other module is not?
Line 106 – Why would you not write (Robledo et al, 2009; Habib 2007; 2011) ?
Line 123-124 – I don’t see why the fact that these bands are used for other mapping applications is relevant
Line 136-7 – Unclear – desire to keep using the tools? This is unclear and can’t possibly be true – the system was not developed b/c users desired to keep using them, the system was developed to deal with detecting mines. – I recommend revising the sentence to: AIDSS was developed to address a need for removing mines quickly in the Republic of Croatia.
Line 151-2 – This sentence should be made more specific. How is it important?
Line 154 – should be “munitions depots”
Line 169 – potential platform, not “platforms”
Line 178 – the difference between a spiral and waterfall methodology is unclear
Line 180 – 190 – the lengthy explanation for excluding radar technology belongs in the discussion perhaps?
Line 224-5 – The fortification objects were defined and described previously
Line 231 – refer to figures and tables indirectly, e.g., which collect information in the visible part of the spectrum, in 3 spectral bands from 239 400 - 700 nm (Table 1).
Line 259 – I would say that this is referred to poorly:
A FLIR Photon 320 (FLIR Systems, Inc. the USA, www.flir.com)
A FLIR Photon 320 (FLIR Systems, Wilsonville, OR, USA)
Line 260 – now? When is now? 2020?
Line 264 – on a computer on the platform – unclear
Line 291 – absolute cinematic mode?
Line 300 – A special GPS device? Maybe just list the type?
Line 352 – what kind of blimp? Why list the types / models for all the other ?
Line 358 – Figure 3a, not 3 a
Line 376 – Why mix use of the terms RPAS and UAV?
Line 393 – Refer to figures and tables indirectly
Line 518 – How about L sub gain instead of L sub (gain)?, etc?
Line 566 – 8 – This does not seem relevant to describe
Line 594-5 – Revise this sentence to refer to the reference indirectly
Line 624, Line 628 – Is this code available? Could you include it as a supplemental file?
Line 635 – But this depends on processing speed of the multi-core system. Is this example relevant?
Line 642 – this module or these modules. Why capitalize, especially if the module is not capitalized consistently.
Line 646 – 669 – This is methods not results.
Figure 10 – The x-axis labels in parts c-e are not readable; the labels are also not logical, eg., could be 5,000 10,000 etc, not 5245….Also the subtitle, which I’m assuming is the name of the file, does not add anything except confusion
Line 704-9 – This is methods
Line 719 – Refer to figures indirectly. 2) This is methods, not results
Line 740 - Refer to figures indirectly.
Line 748 – “the largest recorded terrain” – this is unclear
Line 749-750 – this is methods
Line 753-4 – This is discussion
Line 757- Refer to figures indirectly.
Line 760-2 – This is methods
Line 765, 785 – Figures 13 and 15 can be professionalized by changing the excel defaults, e.g., adding axes, with tick marks,
Line 771,4 – Refer to figures and tables indirectly
Table 3 – What is this value for the subjective confidence of the human observer? Method not described?
Table 790 – Module is capitalized inconsistently. I don’t believe it should be capitalized.
Line 804 – This information on how the images do not need to be orthoreferenced is confusing to me. Don’t the images need to be located in spaces for hazard mapping?
Line 815-816 – The “view of the situation” is vague.
Line 817 – stres – misspelled?
Line 817-8 – How are these habitat mappings relevant?
Line 820 – Isn’t SfM photogrammetry?
Line 834 – The size of the fragmented parts? – this phrase is not clear
Line 834 – Refer to figures and tables indirectly
Line 880 – for the demining community
Author Response
Response to Review Report (Reviewer 1)
The manuscript describes a module for mapping areas of fortification objects (e.g., exploded munitions depots) as well as land mines to develop hazard maps. This manuscript is suitable for publication after revisions. The methods section needs to be reduced and streamlined. There is much extraneous material that is either not needed or could be moved to a supplemental section. One example of this is the SfM section that describes how the metashape software works, or how the geocoding works, which doesn’t appear necessary. Section 2.1 specifically needs to be laid out in a more logical fashion. The section jumps around between the need for the module, how it was developed, and its sections. I recommend the authors start from an outline and re-arrange the section and sentences into a more logical fashion.
The results section is also largely methods, and includes comparison and analysis of the mapping methods that should have be laid out in the methods section. The results section should be laid out to focus on the results of the methods laid out in the methods section.
Lastly, the discussion section is of 1.5 pages in length, and in comparison, with a manuscript of 31 pages, does not appear adequate. The other portions of the paper should be more in line with this length, with perhaps more detailed discussion of the sensor packages relegated to a supplemental section.
There are also too many acronyms in this paper, and especially in its abstract, and include acronyms that are both unnecessary and not commonly used in remote sensing. Please reduce use of acronyms to a core number. It is not possible to keep the number of acronyms in working memory while reading this paper. The citation format is a little odd in that when multiple references are referred to, they are not set off in the same brackets – but scattered around sentences, such as in e.g., Line 106, 119, 125-6, 231-2, etc. These references should be consolidated into one set of brackets.
Author’s Response to Reviewers' Comments:
Thanks to the reviewer for his detailed reading of the text, comments and suggestions. Here are our point-by-point responses. We would like to emphasise that T-AIDSS is a complex system comprising several modules. So far, we have published two articles, in which we described AIDSS as a whole system and producing of mine danger map. In this article, we describe in detail module for data acquisition.
Firstly, we are aware that this is not a 'rocket science' and that there are many more sophisticated and advanced imaging systems. However, the Republic of Croatia faced the problem of mine danger, and in 1998 1,300 square kilometres were contaminated with mines. Croatian scientists (with the help of colleagues from Europe and the world) responded to a call for assistance with humanitarian demining. This research had only a modest budget and went ahead in order to help speed up demining in Croatia. We needed to set up a fairly simple, low-cost system to help reduce SHAs which could be used by Croatian Mine Action Centre personnel. The result of these efforts and research is the AIDSS methodology, which uses the cameras and scientific methods described in this manuscript. This methodology has helped remove 48 kilometres square from the defined SHA, so far. The cost of demining one square meter was about EUR 1.5, so the use of AIDSS helped save over 70 million EUR in Croatia.
The greatest value of AIDSS and data collection module is in adaptability to different platforms, sensors (smaller and modern), close cooperation with users (mine clearance experts), use of various methods for IMP extraction, evaluation of results and repeatability of the above procedures for better results.
Additional line-based suggestions follow.
Ad1 – Reviewer Comment
Line 21-22 – “Display the scene in various ways” is vague and unspecific.
Ad1 - Authors Comment
In our experience, this is neither vague nor unspecific. On the contrary, it is exactly as it is written. Below are some examples to confirm this. We did not see some objects in the images of one sensor, while they could be seen very well in the images of another sensor.
Trenches on a) VNIR DuncanTech image, and b) Photon 320 image
Trenches on a) RGB DuncanTech image, and b) ESAR image.
Ad2 – Reviewer Comment
Line 23-28 – This sentence is not quite grammatical right now, I recommend revising to, “ and some software solutions were also developed.” I do think this sentence could be improved – having all the material in parentheses is not very clear
Ad2 - Authors Comment
We agree with your comment. We rephrased this sentence.
Ad3 – Reviewer Comment
Line 35: revise “(about 10.5% (Human Rights Watch, 1999))” to “(about 10.5%; Human Rights Watch, 1999)
Ad3 - Authors Comment
We rephrased this sentence.
Ad4 – Reviewer Comment
Line 63: sensors
Ad4 - Authors Comment
This has been corrected.
Ad5 – Reviewer Comment
Line 64: numbers less than 10 should be spelled out
Ad5 - Authors Comment
This has been corrected.
Ad6 – Reviewer Comment
Line 73, what the “Impact of Flooding on Mine Action” is, is unclear. Is this a government program? Or a module? If a module why is it capitalized and the other module is not?
Ad6 - Authors Comment
That part of the sentence was mistakenly left in the text. It has been deleted.
Ad7 – Reviewer Comment
Line 106 – Why would you not write (Robledo et al, 2009; Habib 2007; 2011) ?
Ad7 - Authors Comment
This has been corrected.
Ad8 – Reviewer Comment
Line 123-124 – I don’t see why the fact that these bands are used for other mapping applications is relevant
Ad8 - Authors Comment
The name of the manuscript is: The AIDSS Module for Data Acquisition in Crisis Situations and Environmental Protection. The beginnings of the Module's development are closely linked to humanitarian demining, but it has been tested and used for other purposes. Here we want to emphasize the possibility of use in various crisis situations.
Ad9 – Reviewer Comment
Line 136-7 – Unclear – desire to keep using the tools? This is unclear and can’t possibly be true – the system was not developed b/c users desired to keep using them, the system was developed to deal with detecting mines. – I recommend revising the sentence to: AIDSS was developed to address a need for removing mines quickly in the Republic of Croatia.
Ad9 - Authors Comment
We rephrased this sentence.
Ad10 – Reviewer Comment
Line 151-2 – This sentence should be made more specific. How is it important?
Ad10 - Authors Comment
We supplemented the sentence.
Ad11 – Reviewer Comment
Line 154 – should be “munitions depots”
Ad11 - Authors Comment
This has been corrected.
Ad12 – Reviewer Comment
Line 169 – potential platform, not “platforms”
Ad12 - Authors Comment
This has been corrected.
Ad13 – Reviewer Comment
Line 178 – the difference between a spiral and waterfall methodology is unclear
Ad13 - Authors Comment
We have added a sentence for clarification purposes.
Ad14 – Reviewer Comment
Line 180 – 190 – the lengthy explanation for excluding radar technology belongs in the discussion perhaps?
Ad14 - Authors Comment
We agree with your comment. The explanation was moved to Discussion.
Ad15 – Reviewer Comment
Line 224-5 – The fortification objects were defined and described previously
Ad15 - Authors Comment
The sentence has been deleted.
Ad16 – Reviewer Comment
Line 231 – refer to figures and tables indirectly, e.g., which collect information in the visible part of the spectrum, in 3 spectral bands from 239 400 - 700 nm (Table 1).
Ad16 - Authors Comment
We've done that.
Ad17 – Reviewer Comment
Line 259 – I would say that this is referred to poorly:
A FLIR Photon 320 (FLIR Systems, Inc. the USA, www.flir.com)
A FLIR Photon 320 (FLIR Systems, Wilsonville, OR, USA)
Ad17 - Authors Comment
We have provided a better reference.
Ad18 – Reviewer Comment
Line 260 – now? When is now? 2020?
Ad18 - Authors Comment
We specified the time period.
Ad19 – Reviewer Comment
Line 264 – on a computer on the platform – unclear
Ad19 - Authors Comment
We supplemented the sentence.
Ad20 – Reviewer Comment
Line 291 – absolute cinematic mode?
Ad20 - Authors Comment
That confusing sentence was removed.
Ad21 – Reviewer Comment
Line 300 – A special GPS device? Maybe just list the type?
Ad21 - Authors Comment
A translation error has occurred. It's just a separate GPS unit, not a special one.
Ad22 – Reviewer Comment
Line 352 – what kind of blimp? Why list the types / models for all the other ?
Ad22 - Authors Comment
It's a balloon filled with helium. We wanted to show the different platform on which module is already installed.
Ad23 – Reviewer Comment
Line 358 – Figure 3a, not 3 a
Ad23 - Authors Comment
This has been corrected.
Ad24 – Reviewer Comment
Line 376 – Why mix use of the terms RPAS and UAV?
Ad24 - Authors Comment
This has been corrected.
Ad25 – Reviewer Comment
Line 393 – Refer to figures and tables indirectly
Ad25 - Authors Comment
We've done that.
Ad26 – Reviewer Comment
Line 518 – How about L sub gain instead of L sub (gain)?, etc?
Ad26 - Authors Comment
These tags were taken from the reference, so we left them as they are.
Ad27 – Reviewer Comment
Line 566 – 8 – This does not seem relevant to describe
Ad27 - Authors Comment
We agree with your comment. This part of the text has been deleted.
Ad28 – Reviewer Comment
Line 594-5 – Revise this sentence to refer to the reference indirectly
Ad28 - Authors Comment
We've done that.
Ad29 – Reviewer Comment
Line 624, Line 628 – Is this code available? Could you include it as a supplemental file?
Ad29 - Authors Comment
Yes, but, the comments are written in Croatian.
Ad30 – Reviewer Comment
Line 635 – But this depends on processing speed of the multi-core system. Is this example relevant?
Ad30 - Authors Comment
No. The sentence has been deleted.
Ad31 – Reviewer Comment
Line 642 – this module or these modules. Why capitalize, especially if the module is not capitalized consistently.
Ad31 - Authors Comment
This problem is solved.
Ad31 – Reviewer Comment
Ad31 – Reviewer Comment
Line 646 – 669 – This is methods not results.
We partially agree. We have moved to Discusion a part of text describing platform testing, but part of the text about the design and construction of sensor pods not. These are the results of our work on the aerodynamic design of sensor pods, supports for each platform type and all the necessary parts of the data acquisition module. These are the results of the work of our colleague, aeronautical engineer, who designed and built all the sensor pods. The following shows only a small portion of the documentation and images of sensor pods and other platform installation equipment.
Line 704-9 – This is methods
Yes, you're right. However, a brief explanation here is intertwined with the results, so for this reason we have included this text here. It was important for us to show how it affected the quality of the images.
Line 749-750 – this is methods
Yes, you're right. We moved this to section 3.2.
Line 760-2 – This is methods
Yes, you're right. We moved this to section 3.2.
Ad32 – Reviewer Comment
Figure 10 – The x-axis labels in parts c-e are not readable; the labels are also not logical, eg., could be 5,000 10,000 etc, not 5245….Also the subtitle, which I’m assuming is the name of the file, does not add anything except confusion
Ad32 - Authors Comment
We improved the images as much as possible. These are archival images. Divisions are not round figures because we wanted to see the changes in vibrations as better and more accurate as possible.
Ad33 – Reviewer Comment
Line 719 – Refer to figures indirectly.
Line 740 - Refer to figures indirectly.
Line 757- Refer to figures indirectly.
Line 771,4 – Refer to figures and tables indirectly
Line 834 – Refer to figures and tables indirectly
Ad33 - Authors Comment
We've done that.
Ad34 – Reviewer Comment
Line 748 – “the largest recorded terrain” – this is unclear
Ad34 - Authors Comment
We supplemented the sentence.
Ad35 – Reviewer Comment
Line 753-4 – This is discussion
Ad35 - Authors Comment
This is just one sentence, so we left it.
Ad36 – Reviewer Comment
Line 765, 785 – Figures 13 and 15 can be professionalized by changing the excel defaults, e.g., adding axes, with tick marks,
Ad36 - Authors Comment
Of course, this is a matter of presentation. It can always be different.
Ad37 – Reviewer Comment
Table 3 – What is this value for the subjective confidence of the human observer? Method not described?
Ad37 - Authors Comment
This is described in (Krtalić, Bajić, 2019) and (Krtalić, Kuveždić Divjak, Župan, 2019) and section 3.2. In short, subjective confidence is a very important input to the decision support system for producing of mine danger map, which is implemented in the next AIDSS module.
Ad38 – Reviewer Comment
Table 790 – Module is capitalized inconsistently. I don’t believe it should be capitalized.
Ad38 - Authors Comment
The module is part of a larger system and as such we have named it. However, you're right, it's confusing so we've removed all capital letters where they don't need to be.
Ad39 – Reviewer Comment
Line 804 – This information on how the images do not need to be orthoreferenced is confusing to me. Don’t the images need to be located in spaces for hazard mapping?
Ad39 - Authors Comment
If you collect 5000 images of an area and only 55 of them contain indicators, why orthorectify all 5000? After interpreting the original images, only the images that undergo triage are orthorectified and the indicators are mapped. Because our mission is not to produce digital orthophoto, but to detect indicators.
Ad40 – Reviewer Comment
Line 815-816 – The “view of the situation” is vague.
Ad40 - Authors Comment
We supplemented the sentence.
Ad41 – Reviewer Comment
Line 817 – stres – misspelled?
Ad41 - Authors Comment
No, that means: stress (we misspelled, sorry), strong to emphasize vegetation spectral response inside (where the response mixes with mines underground) and outside the minefields (where there are certainly no mines)
Ad41 – Reviewer Comment
Line 817-8 – How are these habitat mappings relevant?
Ad41 - Authors Comment
We remind that the title is: The AIDSS Module for Data Acquisition in Crisis Situations and Environmental Protection, so we have also listed some alternatives to using this module, not just for humanitarian demining.
Ad42 – Reviewer Comment
Line 820 – Isn’t SfM photogrammetry?
Ad42 - Authors Comment
This is the method used in photogrammetry, but certain conditions must be met for its implementation. Rigid pre-treatment must be performed to obtain the best possible photogrammetry results. We have only used this method here without satisfying these stringent conditions. This is emphasized here.
Ad43 – Reviewer Comment
Line 834 – The size of the fragmented parts? – this phrase is not clear
Ad43 - Authors Comment
The SHA in a municipality is not a compact surface. It consists of several parts. Which platform will be used to collect the images depends on the size and scatter of those parts. This is example of SHA (red poligons) in one municipality.
Ad44 – Reviewer Comment
Line 880 – for the demining community
Ad43 - Authors Comment
We've done that.
Best regards
Andrja Krtalić
Milan Bajić
Tamara Ivelja
Ivan Racetin

Reviewer 2 Report
This paper reports the outcomes of a series of Framework grants and as such appears outdated in a number of respects. Many of the references are quite old, but that is because the work has been undertaken over a period of ten or more years. The issue of identifying mine fields after military action has finished is of major importance because of the risks that they offer to civilians, both in terms of deaths and serious injuries. Anything that helps identify and clear them so that the land can be returned to productive use is of major benefit to both the local residents and society at large. The resources deployed in terms of platforms for mounting equipment and analysing the results are well described, together for the reasons for choosing the eventual configurations. Whilst equipment is always improving (I would look for more modern and smaller cameras for example if starting a similar project now) this can only be an advantage as the size and weight of the equipment cases would hopefully be a lot less. The results reported and the conclusions would make setting up such a project a lot quicker as much trial and error would be avoided. I hope that the software (for example the Matlab functions) are made available for future use.
The English is generally excellent and the style is highly readable. There did however seem to be some problems around lines 640-650 where it almost became unreadable (at least for me). This needs to be checked to make sure that the text matches the sense that the authors intend.
Author Response
Response to Review Report (Reviewer 2)
Top of Form
This paper reports the outcomes of a series of Framework grants and as such appears outdated in a number of respects. Many of the references are quite old, but that is because the work has been undertaken over a period of ten or more years. The issue of identifying mine fields after military action has finished is of major importance because of the risks that they offer to civilians, both in terms of deaths and serious injuries. Anything that helps identify and clear them so that the land can be returned to productive use is of major benefit to both the local residents and society at large. The resources deployed in terms of platforms for mounting equipment and analysing the results are well described, together for the reasons for choosing the eventual configurations. Whilst equipment is always improving (I would look for more modern and smaller cameras for example if starting a similar project now) this can only be an advantage as the size and weight of the equipment cases would hopefully be a lot less. The results reported and the conclusions would make setting up such a project a lot quicker as much trial and error would be avoided. I hope that the software (for example the Matlab functions) are made available for future use.
The English is generally excellent and the style is highly readable. There did however seem to be some problems around lines 640-650 where it almost became unreadable (at least for me). This needs to be checked to make sure that the text matches the sense that the authors intend.
Author’s Response to Reviewers' Comments:
Thanks to the reviewer for his detailed reading of the text, friendly comments and suggestions. We would like to emphasise that T-AIDSS is a complex system comprising several modules. So far, we have published two articles, in which we described AIDSS as a whole system and producing of mine danger map. In this article, we describe in detail module for data acquisition.
Firstly, we are aware that this is not a 'rocket science' and that there are many more sophisticated and advanced imaging systems. However, the Republic of Croatia faced the problem of mine danger, and in 1998 1,300 square kilometres were contaminated with mines. Croatian scientists (with the help of colleagues from Europe and the world) responded to a call for assistance with humanitarian demining. This research had only a modest budget and went ahead in order to help speed up demining in Croatia. We needed to set up a fairly simple, low-cost system to help reduce SHAs which could be used by Croatian Mine Action Centre personnel. The result of these efforts and research is the AIDSS methodology, which uses the cameras and scientific methods described in this manuscript. This methodology has helped remove 48 kilometres square from the defined SHA, so far. The cost of demining one square meter was about EUR 1.5, so the use of AIDSS helped save over 70 million EUR in Croatia.
The greatest value of AIDSS and data collection module is in adaptability to different platforms, close cooperation with users (mine clearance experts), use of various methods for IMP extraction, evaluation of results and repeatability of the above procedures for better results.
Ad1 – Reviewer Comment
There did however seem to be some problems around lines 640-650 where it almost became unreadable (at least for me). This needs to be checked to make sure that the text matches the sense that the authors intend.
Ad1 - Authors Comment
We removed the confusing and unnecessary text.
Best regards
Andrja Krtalić
Milan Bajić
Tamara Ivelja
Ivan Racetin

Reviewer 3 Report
The presented article is a very valuable and important resource of information. It contains data on innovative and modern solutions for multi-sensors data acquisition worth publishing. I am impressed by the vast amount of research work and rich experience of the authors. Unfortunately, in my opinion, in its current form the article is not a formal scientific work. Although the text has been arranged in the IMRAD structure, it is in fact a very valuable synthesis of the technical report. Therefore, I recommend rejecting the manuscript.
Please notice that I will suggest Editors to consider publishing this material as a Technical Report, then the IMRAD structure does not have to be maintained.
Author Response
Response to Review Report (Reviewer 3)
Top of Form
The presented article is a very valuable and important resource of information. It contains data on innovative and modern solutions for multi-sensors data acquisition worth publishing. I am impressed by the vast amount of research work and rich experience of the authors. Unfortunately, in my opinion, in its current form the article is not a formal scientific work. Although the text has been arranged in the IMRAD structure, it is in fact a very valuable synthesis of the technical report. Therefore, I recommend rejecting the manuscript.
Please notice that I will suggest Editors to consider publishing this material as a Technical Report, then the IMRAD structure does not have to be maintained.
Author’s Response to Reviewers' Comments:
Thanks to the reviewer for his detailed reading of the text, comments and suggestions. We would like to emphasise that T-AIDSS is a complex system comprising several modules. So far, we have published two articles, in which we described AIDSS as a whole system and producing of mine danger map. In this article, we describe in detail module for data acquisition.
Firstly, we are aware that this is not a 'rocket science' and that there are many more sophisticated and advanced imaging systems. However, the Republic of Croatia faced the problem of mine danger, and in 1998 1,300 square kilometres were contaminated with mines. Croatian scientists (with the help of colleagues from Europe and the world) responded to a call for assistance with humanitarian demining. This research had only a modest budget and went ahead in order to help speed up demining in Croatia. We needed to set up a fairly simple, low-cost system to help reduce SHAs which could be used by Croatian Mine Action Centre personnel. The result of these efforts and research is the AIDSS methodology, which uses the cameras and scientific methods described in this manuscript. This methodology has helped remove 48 kilometres square from the defined SHA, so far. The cost of demining one square meter was about EUR 1.5, so the use of AIDSS helped save over 70 million EUR in Croatia.
The greatest value of AIDSS and data collection module is in adaptability to different platforms, sensors (smaller and modern), close cooperation with users (mine clearance experts), use of various methods for IMP extraction, evaluation of results and repeatability of the above procedures for better results.
Best regards
Andrja Krtalić
Milan Bajić
Tamara Ivelja
Ivan Racetin

Round 2
Reviewer 3 Report
I accept the Auhors' explanations.